# Construct and criterion validity of the HEXACO Medium School Inventory Extended (MSI-E)

**Francesca Mottola**[1], **Lucia Abbamonte**[1], **Lucia Ariemma**[1], **Augusto Gnisci**[1]*,
**Roberto Marcone**[1], **Andrea Millefiorini**[1], **Marco Perugini**[2], **Vincenzo Paolo Senese**[1],
**Ida Sergi**[1]

**1** Department of Psychology, University of Campania "Luigi Vanvitelli", Caserta, Italy, **2** Department of Psychology, University of Milan-Bicocca, Milan, Italy

* augusto.gnisci@unicampania.it

## Abstract

In this cross-sectional study we aimed at: 1) validating the observer (Obs) version of HEXACO Medium School Inventory Extended (MSI-E); 2) establishing convergent and divergent construct validity of the HEXACO-MSI-E; 3) establishing criterion validity of HEXACO-MSI-E. We administered the HEXACO-MSI-E, the Big Five Questionnaire-Children (BFQ-C), the Internalizing and Externalizing scales of Youth Self Report (YSR), some items of the 2019 Middle School Youth Risk Behavior Survey (YRB) and some items about adolescents' values, beliefs, behaviors, and desired features of a possible future job to 1175 adolescents ($M_{age}$ = 12.03, SD = 0.89) and the observer version of these measures (except for BFQ-C) to 854 parents or legal guardians. The factorial structure and the reliability of the Observer Report of HEXACO-MSI-E were confirmed. Convergent and divergent validity were successfully established with a version of the inventory filled out by parents. Convergent and divergent validity were also established with BFQ-C notwithstanding some only apparent anomalies. Criterion validity was established with respect to four specific groups of criteria collected in both self-report and observer form. HEXACO-MSI-E traits were more predictive with respect to self-report than to observer criteria and the majority were common. Together with the positive results of this study, implications and issues for future studies are discussed.

## Introduction

During the past two decades, there emerged a near-consensus that human personality variation can be summarized by five broad dimensions that are collectively known as the Big Five [1–3]. These five factors have been labelled (I) Extraversion, (II) Agreeableness, (III) Conscientiousness, (IV) Neuroticism, and (V) Intellect/Imagination (or Openness to Experience). The Big Five factors were repeatedly recovered in early lexical studies of personality structure based on the personality-descriptive adjectives of the English language [1, 2, 4–6] and have since been widely popularized in the questionnaire-based Five-Factor Model or FFM [7, 8]. The FFM is sometimes identified as OCEAN from the initials of the factors [7]. In recent years, the lexical approach to personality structure has been extended to several languages other than

**Funding:** The project EMERGE (ID 389) has been funded by the programme V:ALERE 2019 of the University of Campania "Luigi Vanvitelli" (D.R. 906 del 4/10/2019, prot. n. 157264, 17/10/2019). The funders had no role in study design, data collection and analysis, decision to publish, or preparation of the manuscript. https://www.psicologia. unicampania.it/home-emerge/27-emerge/1558-home-emerge-en.

**Competing interests:** The authors have declared that no competing interests exist.

English. Across 'standard' lexical studies—that is, investigations characterized by analyses of self- or peer ratings on personality-descriptive adjectives only—a surprisingly consistent result has been the emergence of a six-factor solution [9]. This structure has been recovered from re-analyses of data from languages such as Dutch [10], French [11], German [12, 13], Hungarian [14, 15], Italian [16–18], Korean [19], and Polish [20]. Lee and Ashton [21] named the lexically derived six-dimensional model of personality as HEXACO model, using an acronym derived from the names of the six factors: Honesty-Humility (H), Emotionality (E), eXtraversion (X), Agreeableness (A), Conscientiousness (C), and Openness to Experience (O). Three of these six cross-language factors are very similar to the Big Five equivalents, Extraversion, Conscientiousness, Intellect/Imagination dimensions. Two other factors roughly represent rotated variants of the Big Five Agreeableness and Emotional Stability. The Agreeableness factor blends the gentleness of Big Five Agreeableness with the even temper of Big Five Emotional Stability, whereas the Emotionality factor blends the vulnerability of (low) Big Five Emotional Stability with the sentimentality of Big Five Agreeableness. Finally, the remaining factor of this six-dimensional structure, Honesty-Humility, has no equivalent in the previous model and is defined by contents suggestive of honesty and humility (e.g., sincere, fair, unassuming vs. sly, greedy, pretentious). The recovery of this six-dimensional structure from the indigenous and representative personality descriptors of diverse languages establishes this solution as the best available summary of the domain of human personality dispositions. In addition to its empirical accuracy, the HEXACO framework also has some advantages in terms of theoretical interpretability. Indeed, the Honesty–Humility and Agreeableness factors correspond to two complementary aspects of reciprocal altruism, whereas the Emotionality factor corresponds to kin altruistic tendencies [22]. Furthermore, the Extraversion, Conscientiousness, and Openness to Experience factors can be viewed as dimensions of social engagement, task-related engagement, and idea-related engagement, respectively [23].

Research suggests that the HEXACO model has practical and theoretical advantages over the Big Five for predicting individually, organizationally, and academically relevant outcomes [22, 24]. For instance, Honesty-Humility captures facets of personality (e.g., Sincerity, Modesty) that are not well represented in Big Five scales [23]. Literature [25–27] has shown that Honesty-Humility appears to be a good predictor of antisocial, manipulative, and unethical behaviors such as bullying, sexual harassment, premeditated revenge, psychopathy, narcissism, and Machiavellianism. Other research has investigated this personality factor in adolescent populations, demonstrating that low levels of Honesty-Humility are a strong predictor of bullying behaviors, even more so than other personality factors, such as low Agreeableness [25, 28]. Moreover, Honesty-Humility was negatively correlated with symptoms of the oppositional-defiant disorder (ODD) and conduct disorder (CD) [29]. Research on the HEXACO and bullying demonstrate that Honesty-Humility is typically the best predictor of bullying behavior in Western adolescents [25, 30]. Similar pattern emerged from a study conducted by Book et al. [25]. They found a significant negative correlation in traditional bullying with Honesty-Humility, Emotionality, Agreeableness, and Conscientiousness traits. Moreover, Honesty-Humility and Conscientiousness predicted cyberbully perpetration, whereas Emotionality, Extraversion, and Conscientiousness predicted cyberbully victimization. Book and colleagues [31] showed that Honesty-Humility was negatively associated with self-reports of both reactive and proactive aggression, instead, Agreeableness with reactive aggression. Moreover, a study about problematic smartphone use has shown that high levels of Emotionality and low levels of Honesty-Humility, Conscientiousness, eXtraversion, and, to a lesser extent, of Openness to Experience, predicted problematic smartphone use in adolescents [32]. Weller and Tikir [33] have investigated the associations between all HEXACO personality traits and domain specific risk-taking. They found that while taking risks in

social and recreational domains is associated with high Openness to Experience, taking risks in ethical and health/safety domains is associated with low Honesty-Humility. In most studies, the influence of Honesty-Humility on externalizing problems remained statistically significant when controlling for the other HEXACO traits [25, 30, 34], with Agreeableness sometimes also making a significant negative contribution. As far as adolescents' beliefs is concerned, Aghababaei and colleagues [35] found that Honesty–Humility factor was one of the strongest personality correlates of religiosity. Higher scores on religiousness were also associated with Agreeableness, Conscientiousness and, to some extent, Extraversion.

In conclusion, this brief review of the predictive capacity of the HEXACO with respect to many criteria together with many other systematic review on the topic [36–41] shows that personality traits exert their predictive effects on a vast array of emotional (e.g., anxiety, depression, withdraw, somatization) and behavioral (e.g., aggressive, delinquency) problems, of risk factors (e.g., being bullied, consumption of alcohol, smoke or drugs, eating and sleeping habits, TV or smartphone use, safety measures), of personal passions (e.g., music, art, sports, writing, science), values and beliefs (e.g., believing in God, loving animals, being a volunteer), and different behaviors (e.g., being respectful, making jokes, desiring a determined job). We should add that many authors point out the issue that each criterion can be evaluated by the subjects themselves or by significant others, as when peers, partners, parents, etc., are asked to rate the individual. In this case, the 'method' with which each criterion is evaluated is also important because the prediction through the traits may change depending on the method itself [38–40].

As regards the assessment of personality in children, in literature, temperament-based measures have been used to assess individual differences among infants and children, whereas personality-based measures have been used to assess differences among adults [42]. Currently, there is a large body of literature that has evaluated the personality in children which shows a clear conceptual overlap between the traits identified in temperament research and the traits assessed in personality research. As argued by Shiner [43], it might be reasonable to assert that temperament and personality should be seen as the same basic set of traits, the former manifested early in life and the latter manifested later in life [44–46]. In the extant literature, all the studies carried out on children have considered the FFM as the theoretical framework of references [47–50], showing that by the age of 10 to 12 the five dimensions of personality are already recognizable. Longitudinal studies have shown the reliability and validity of the FFM for adolescents aged 12–18 years [51] and for children aged 4–12 years [52]. As it is clear, all the studies just mentioned on children have used the five-factor model, and the most commonly used self-report measure for children and early adolescents (from 8 to 12 years) was the Big Five Questionnaire–Children or BFQ-C [53]. Only recently, Farrell and colleagues [28] have adopted the HEXACO model for investigating personality in adolescents (age 11–17). The results showed that using the HEXACO model gives unique and meaningful contributions to the study of adolescents' behaviors, indicating that Honesty-Humility is specifically related to antisocial behaviors [54]. However, the authors used the adult version of the inventory rather than adapting the content of the items to the adolescent's age. For this reason, more recently, some authors have developed tailored inventories for children/adolescents according to the HEXACO model [55, 56]. In particular, Sergi and colleagues [55] have developed and validated a preliminary 48-item version of a HEXACO inventory adapted for 10–14 years old adolescents, named HEXACO-Middle School Inventory (HEXACO-MSI). This inventory has been shown to have adequate psychometric properties and provided important results about many aspects of validity, such as dimensional validity, internal consistency, 1-month stability, invariance across sex and classes (7th, 8th and 9th grade), construct validity with respect to an observer report and to BFQ-C and,

finally, criterion validity with respect to school grades. Indeed, three factors, Conscientiousness, Honesty-Humility, and eXtraversion proved good predictors of school grades. Despite several merits, this preliminary version of the inventory had some weaknesses regarding the extent of content validity and the balance among reversed and not reversed items. To solve these problems, Gnisci and colleagues set up two studies [56] for developing a new extended version of the inventory, more representative of the facets belonging to each dimension according to the reference framework. The novel inventory of 192 items is called HEXACO--Medium School Inventory Extended (HEXACO-MSI-E), for which have already just proven, together with content validity, some basic psychometric qualities: the balance of reversed items in each facet and thus in each factor, the emergence of six factors according to the HEXACO model (dimensional validity), the invariance of such structure across gender and classes of medium school (cross-validation), the reliability as internal consistency and as temporal stability. In particular, with both cross-sectional invariance (across 7th, 8th, and 9th grade) and longitudinal reliability (1-year stability) methods, this study demonstrated that the inventory measures the six personality traits that were well structured across ages and stable across time.

In the present study, to further validate the HEXACO-MSI-E, we provide additional analyses on the data collected in the Gnisci et al. [56] second study but not yet published, considering new and different variables and inventories, and we include a new sample of participants (parents or legal guardians) on which we measured observer variables. The general aim of this contribution is to check the construct and criterion validity of the HEXACO-MSI-E inventory in an Italian sample of adolescents. To reach this aim, we will:

1. validate the observer (Obs) version of HEXACO-MSI-E. Given that observer reports from parents, caretakers, legal guardians are often used particularly in child and adolescent psychology, making available a validated observer report of the HEXACO-MSI-E form seems useful. In particular, we want: a) to verify the six-dimensional structure of personality in adolescence coherent with the HEXACO model, when the inventory is filled out by a significant other of the adolescent (one of the parents or a legal guardian), with confirmatory factorial analysis techniques (dimensional validity of the Observer Report); b) to provide evidence that each trait of adolescents, measured by the observer, is reliable by considering Cronbach's alpha coefficient (internal consistency of the Observer Report);

2. establish convergent and divergent construct validity of the HEXACO-MSI-E. In particular, we will contrast the inventory with: a) its Observer form; and b) the Big Five inventory for adolescents validated in Italy, that is, the BFQ-C [53]. The BFQ-C has approximately a similar behavioral domain of HEXACO-MSI-E but rather different contents depending on each trait. Therefore, a priori, we do not expect the traditional results of a multitrait-multimethod matrix whereby the correlations between the 'same' traits have to be high and higher than correlations between different traits.

3. establish criterion validity with respect to several criteria, collected in both self-report and observer form, falling in one of these four categories: a) behavioral and emotional problems of adolescents, particularly, internalizing- and externalizing-related problems, as measured by the Youth Self-Report inventory (YSR) [57]; b) risk factors in adolescence, as measured by Youth Risk Behavior survey (YRB) [58]; c) passions, values, beliefs and behaviors of adolescents (PVB&B) [38–40], that explored many aspects of the adolescents preferences (e.g., religion, animal loving, music, being engaged, science and art, respect, sleeping, making jokes, feeling good); d) and, finally, the Features of a Future Job (FFJ) the participants will do when they grow up (inspired by Soto) [40].

## Methods

### Participants and procedure

**Participants.** The total participants to this research were 2.029, of which 1175 were middle school adolescents (52.2% female; Mean age = 12.03, SD = 0.89, age range 10–14) and 854 parents or legal guardians. All the adolescents were administered the HEXACO-MSI-E and a subset of specific instruments/variables, depending on the partaking to one of three randomly determined different subsampling (Sample 1, 2, and 3) as specified below.

Sample 1, formed by 400 adolescents (52.8% female; Mean age = 12.01, SD = 0.89), was administered the BFQ-C. Sample 2, formed by 388 adolescents (53.4% female; Mean age = 12.07, SD = 0.91), was administered a subset of specific criterion variables related to internalizing and externalizing problems of adolescents (YSR). Sample 3, formed by 385 adolescents (50.6% female; Mean age = 12.01, SD = 0.88), was administered with three subsets of specific criterion variables related to risk factors in adolescence (YRB), personal passions, values, beliefs, and behaviors (PVB&B), and desired features of a possible future job (FFJ).

The samples of 854 parents or legal guardians was divided into two random subsamples. The first consisted of 403 parents/legal guardians who provided observer reports of the same items of the HEXACO-MSI-E but of course in the third person (Obs HEXACO-MSI-E). The second consisted of 451 participants who provided observer reports on the four subsets of specific criterion variables also administered to their children, in the third person as well.

**Recruitment procedure and informed consent.** First, the project received approval from the local Ethics Committee of the Department (approval number 13/26.05.2020). Then, using a convenience sampling technique, we reached the available schools (November 2020) and the research plan was approved by the Directors of the schools and by their Councils. Third, parents and adolescents were informed about the project by the school, by the research assistants, by written instructions and by video- and audio-recordings, specially prepared. Fourth, once informed, the parents/legal guardians were administered the online protocol, at the beginning of which they read the basic information regarding the research and then if they wanted, provided the authorization to the participation for their children and themselves. Fifth, at the beginning of their online protocol, the adolescents read a written description of the research and were asked about their willingness to participate in the research. It was specified that the responses were anonymous, and data were treated collectively. The protocol filled out by parents and their children was associated with an alphanumeric code; therefore, the authors never had access to information that could identify participants. The children and their parents were informed that they were free to decline to take part in the data collection at any time and without any consequence.

**Data collection.** Data were collected between April and May 2021. Participants completed online protocol on Qualtrics platform. Due to the Covid-19 and according to the ministerial indications, the involved schools provided distance education, using online platforms. For this reason, adolescents were administered the online protocol in their virtual classrooms by research assistants and in the presence of the teacher. The involved parents were previously contacted and instructed by research assistants, who, also, sent them the link to the protocol. They were asked to fill out the online protocol in a few days. It was also specified that only one parent or legal guardian had to complete the protocol.

## Measures

**Demographic information.**   At the beginning of the protocol, basic information such as gender, age, class, and information on parents (i.e., educational level and professional status) were requested.

**The self-report and observer HEXACO-MSI-E.**   We administered a preliminary form of the HEXACO-MSI-E consisting of 219 items, then we selected the best 192 items (6 per each facet with reverse balanced), as detailed in Gnisci et al. [56], to assess the six personality dimensions of the HEXACO model in a self-report form. The adolescents rated their own behavior on a 5-point Likert scale ranging from 1 (*True*) to 5 (*False*). We used the same exact inventory but worded in the third person to have an evaluation reported by a significant other (parent or legal guardian). The HEXACO-MSI-E is available in self-report and observer report forms (Obs HEXACO-MSI-E), in both a female and male version at the webpage: https://www. psicologia.unicampania.it/strumenti-di-misura. Hereafter, we will refer to the HEXACO traits with the first letter of the dimension.

**Big Five Questionnaire-Children.**   The BFQ-C-Short Form [53] is a self-report measure composed by 30 items that assesses the personality using Five Factor Model (Extraversion, Agreeableness, Openness, Conscientiousness, and Neuroticism). In order to make data collection uniform, we have used the same 5-point Likert scale used for the HEXACO-MSI-E. We carried out a five-factor Principal Component Analysis (PCA). The five-factor solution accounted for 53.96% of the total variance. The oblimin rotated solution showed that all items had an adequate loading on the single, pertinent factor, that no item (except one) cross-loaded on a not pertinent factor and that the five factor were modestly related ($rs < |.32|$). The only item that loaded on two different factors with a value higher than .30 was item 4 ("I treat my peers with affection") on X (.41) and C (.34), which anyway loaded also on the pertinent factor (A) even if with a slightly lower loading (.33). All factors had good levels of internal consistency, as shown by alphas (OCEAN): $\alpha = .83$, $\alpha = .80$, $\alpha = .78$, $\alpha = .74$ and $\alpha = .84$, respectively.

Now we will describe the four groups of items used for establishing criterion validity. The English version of each item, in both self-report and observer form, can be read in the respective tables of the results.

**Youth self and observer report.**   The YSR [57] is a widely used instrument for assessing emotional (internalizing) and behavioral (externalizing) problems. In literature it is known as YSR, in this contribution, we will label the observer form as Youth Observer Report (YOR) for avoiding the oxymoronic expression 'Observer Youth Self Report'. In total, this inventory contains 112 items with which the participants had to rate to what extent they were applicable to them (1 = *Not true*, 2 = *Somewhat true*, and 3 = *Very true*). For the purposes of this study, to get a measure of internalizing and externalizing problems, we considered the following subscales: Aggressive Behavior (16 items), Delinquent Behavior (11 items), Anxious/Depressed (10 items), Somatic Complaints (3 items), and Withdrawn/Depressed (7 items). Items examples: "I feel lonely"; "I cry a lot"; "I steal at home".

**Self-report and observer middle school Youth Risk Behavior survey.**   The 2019 Middle School Youth Risk Behavior Survey (hereafter YRB) [58] is a 49-item measure that aims to monitor health risk behaviors that contribute to the leading causes of death, disease, injury, and social problems among youth, such as nutrition, weight status, tobacco use, alcohol, and other drug use. We selected the 16 items from the original inventory that we considered appropriate for the adolescents, sometime providing light adaptation in the wording of the sentences (for example: "How often do you wear a seat belt when riding in a car?"; "Have you ever ridden in a car driven by someone who had been drinking alcohol?") and administered them to adolescents. Adolescents rated each behavior using different response scales (e.g., *Yes/No; Never/*

*Rarely/Sometimes/Almost always/Always*). The YRB was administered also in an observer form to parents/legal guardians (Obs YRB).

**Self-report and observer personal passions, values, beliefs and behaviors of adolescents.** To get information about adolescents' personal passions, values, beliefs, and behaviors (hereafter we will call it PVB&B), a questionnaire was devised ad hoc and administered to them. The questions were selected from criteria typically associated with FFM and HEXACO models in adult personality [36–40] and adapted for adolescence when necessary. The final version of questionnaire consisted of 18 items (items example "I listen to older people with respect"; "I believe in God"; "When there is no school, I sleep late regardless of when I go to bed"). The adolescents rated each item using different response scales (e.g., *Yes/No; Never/Rarely/Sometimes/Often/Always*). This list of items was administered also in an observer form (Obs PVB&B).

**Self-report and observer desired features of a possible future job.** To get information about adolescents' desired Features of a possible Future Job (FFJ), an inventory, consisting of 7 items (for example: "It is important for you that the job is honest"; "...that the job is organized and planned") was built ad hoc taking inspiration by Soto [40], but rethought and adapted to adolescents. The adolescents rated each item using a 5-point Likert scale ranging from 1 (*Not important*) to 5 (*Extremely important*). These items were administrated also in an observer form.

## Data analysis

Preliminarily, the data were analyzed to identify any distribution problems of responses or missing cases. Subsequently, analyses were performed to verify the psychometric properties of the observed version (Obs HEXACO-MSI-E) of the inventory, and the validity of the HEXACO-MSI-E inventory. To verify the latent structure of the observer inventory (Obs HEXACO-MSI-E), a confirmatory factorial analysis was conducted on the 24 facets related to the six factors. Then, Cronbach's alpha was calculated for each dimension and facet to verify the reliability of the inventory.

To verify the construct validity of the inventory, considering both convergent and divergent components, a correlation analysis was performed between the six traits of the HEXACO-MSI-E, the six traits of the Obs HEXACO-MSI-E, and the five traits of the BFQ-C. The results obtained were entered into a multitrait-multimethod matrix (MTMM) to investigate the nomological validity of the inventory.

For establishing the criterion validity, we first created a dummy variable for each dichotomous criterion, then we correlated each self-report HEXACO-MSI-E trait with each criterion (see S1 File for Supplementary Materials) and finally we regressed (Method Stepwise) each criterion on the predictors (i.e., the six self-report traits). The criteria were grouped in four groups (see above). We corrected the significance of the betas for multiple testing within each group of criteria with a False Discovery Rate method (FDR) [59]. Correlations between self-report and observer criteria were also calculated. Here we will present the minimum and the maximum correlation within each group of criteria (complete Tables in S1 File).

## Results

### Dimensionality and reliability of HEXACO-MSI-E observer form

**Confirmatory factor analysis on facets.** CFA on facets shows that the 6-factor model had sufficient fit indices. The analysis of the modification indices revealed the significance of some additional parameters. Accordingly, two cross-loadings were considered, RMSEA = .072; 90% CI [.066, .078]; CFI = .929, DWLS$X^2$(235, N = 403) = 723.66, p < .001. Table 1

**Table 1. Descriptive statistics, factor loadings of the 6-factor CFA on the facets of the Obs HEXACO-MSI-E and reliability of its dimensions and facets (N = 403).**

| Factor / Facet | M (SD) | Skewness | Kurtosis | H | E | X | A | C | O | Reliability alpha |
|---|---|---|---|---|---|---|---|---|---|---|
| Honesty-Humility | 4.09 (.54) | -.70 | .32 | | | | | | | .89 |
| Fairness | 4.59 (.53) | -1.90 | 4.95 | .737 | | | | | | .72 |
| Greed Avoidance | 3.52 (.86) | -.31 | -.55 | .605 | | | | | | .81 |
| Modesty | 4.04 (.74) | -.84 | .47 | .645 | | | | | | .75 |
| Sincerity | 4.22 (.61) | -.73 | -.12 | .806 | | | | | | .66 |
| Emotionality | 3.55 (.57) | -.25 | .24 | | | | | | | .88 |
| Anxiety | 3.73 (.60) | -.38 | .40 | | .636 | | | | | .58 |
| Dependence | 3.35 (.84) | -.12 | -.20 | | .727 | | | -.371 | | .85 |
| Fearfulness | 3.34 (.82) | -.26 | -.11 | | .579 | | | | | .75 |
| Sentimentality | 3.77 (.82) | -.48 | -.28 | | .791 | .402 | | | | .81 |
| Extraversion | 3.83 (.67) | -.61 | -.15 | | | | | | | .93 |
| Social Boldness | 3.44 (.90) | -.34 | -.55 | | | .735 | | | | .85 |
| Sociability | 4.11 (.79) | -1.02 | .45 | | | .638 | | | | .85 |
| Liveliness | 3.90 (.81) | -.83 | .15 | | | .888 | | | | .85 |
| Social Self-Esteem | 3.86 (.73) | -.44 | -.54 | | | .773 | | | | .81 |
| Agreeableness | 3.52 (.67) | -.34 | -.16 | | | | | | | .92 |
| Flexibility | 3.38 (.71) | -.48 | .28 | | | | .707 | | | .73 |
| Forgivingness | 3.55 (.87) | -.36 | -.43 | | | | .613 | | | .87 |
| Gentleness | 3.95 (.68) | -.77 | .94 | | | | .839 | | | .77 |
| Patience | 3.19 (1.02) | -.23 | -.85 | | | | .810 | | | .89 |
| Conscientiousness | 3.33 (.82) | -.17 | -.71 | | | | | | | .95 |
| Diligence | 3.69 (.99) | -.43 | -.72 | | | | | .884 | | .91 |
| Organization | 2.83 (1.10) | .30 | -.93 | | | | | .470 | | .91 |
| Perfectionism | 3.38 (1.04) | -.16 | -.93 | | | | | .882 | | .91 |
| Prudence | 3.42 (.84) | -.38 | -.01 | | | | | .802 | | .82 |
| Openness to Experience | 3.39 (.69) | -.28 | -.17 | | | | | | | .91 |
| Aesthetic Appreciation | 3.13 (1.01) | -.22 | -.73 | | | | | | .831 | .88 |
| Inquisitiveness | 3.30 (.97) | -.31 | -.65 | | | | | | .799 | .85 |
| Creativity | 3.78 (.84) | -.71 | .16 | | | | | | .666 | .82 |
| Unconventionality | 3.37 (.65) | .02 | -.26 | | | | | | .476 | .63 |

*Note.* H = Honesty/Humility; E = Emotionality; X = Extraversion; A = Agreeableness; C = Conscientiousness;
O = Openness to Experience.

shows the standardized factor loadings of the final model. The facets that loaded on a different factor with a value higher than .30 were Sentimentality of E on X, and Dependence of E on low C. However, the two facets had much stronger loadings on the pertinent factors.

As shown in Table A in S1 File, correlations for the latent factors ranged from |.033| to |.616| with a mean of |.311|. Relevant correlations (> |.40|) were observed between C and O (.616), H and A (.606), H and E (.436), E and X (-.433) and H and C (.409).

**Reliability.** All dimensions of Obs HEXACO-MSI-E had at least excellent levels of internal consistency, as shown by alpha indexes reported in the Table 1 (range = .88-.95). As for the facets, 3 values were greater than .90, 13 ranged between .80 and .90, 5 between .70 and .79, 2 between .60 and .69, and only 1 below .60 (.58).

**Distribution.** Skewness and kurtosis provide evidence of normality for each factor (Table 1). Note, however, that one facet (Fairness) was more skewed than the other facets (i.e., asymmetric toward high scores) and had a higher kurtosis (> 3).

## Convergent and divergent construct validity

The multitrait–multimethod matrix of self-report HEXACO-MSI-E (N = 1175), Obs HEXA-CO-MSI-E (N = 343) and the BFQ-C (N = 400), reporting observed (above the diagonal) and latent traits (below) correlations is shown in Table 2. For the interpretation, we will focus on correlations between latent factors.

As regard self-report and Obs HEXACO-MSI-E, first, correlations between the same self-report and observer traits of HEXACO-MSI-E showed a very good convergent validity: homo-trait-heteromethod correlations ranged from .529 to .705. Second, correlations between different self-report and observer traits–that is, heterotrait-heteromethod coefficients–were, in general, low (max r = |.372|) and always lower than homotrait-heteromethod coefficients. Therefore, convergent and divergent construct validity of HEXACO-MSI-E with respect to the observer form of HEXACO-MSI was established.

**Table 2. MTMM matrix for convergent and discriminant validity of the HEXACO-MSI-E with the Obs HEXACO-MSI-E and the BFQ-C.**

| | Scale | 1^ | 2 | 3 | 4 | 5 | 6 | 7 | 8 | 9 | 10 | 11 | 12 | 13 | 14 | 15 | 16 | 17 |
|---|---|---|---|---|---|---|---|---|---|---|---|---|---|---|---|---|---|---|
| | HEXACO MSI-E | | | | | | | | | | | | | | | | | |
| 1 | H | **.994**\* | .283\* | .099\* | .556\* | .371\* | .194\* | .539\* | .253\* | .069 | .330\* | .264\* | .155\* | -.405\* | .095 | .414\* | .337\* | .160\* |
| 2 | E | .284\* | **.995**\* | -.179\* | .175\* | .126\* | .100\* | .285\* | .556\* | -.085 | .153\* | .089 | .096 | .022 | -.026 | .226\* | .172\* | -.092 |
| 3 | X | .113\* | -.157\* | **.996**\* | .341\* | .305\* | .200\* | .008 | -.169\* | .542\* | .239\* | .127\* | .026 | -.445\* | .725\* | .356\* | .302\* | .403\* |
| 4 | A | .556\* | .172\* | .368\* | **.996**\* | .385\* | .203\* | .382\* | .152\* | .238\* | .600\* | .221\* | .135\* | -.705\* | .265\* | .736\* | .479\* | .241\* |
| 5 | C | .381\* | .113\* | .300\* | .384\* | **.994**\* | .517\* | .217\* | .084 | .222\* | .183\* | .691\* | .362\* | -.465\* | .263\* | .356\* | .802\* | .529\* |
| 6 | O | .264\* | .115\* | .192\* | .278\* | .540\* | **.983**\* | .113\* | .012 | .201\* | .163\* | .320\* | .673\* | -.232\* | .206\* | .237\* | .385\* | .459\* |
| | HEXACO MSI-E Obs | | | | | | | | | | | | | | | | | |
| 7 | Hob | .533\* | .264\* | .022 | .367\* | .207\* | .152\* | **.995**\* | .318\* | .034 | .506\* | .296\* | .084 | -.419\* | -.048 | .368\* | .313\* | .109 |
| 8 | Eob | .230\* | .529\* | -.152\* | .134\* | .064 | .022 | .275\* | **.986**\* | -.269\* | .206\* | .000 | -.022 | -.092 | -.089 | .132 | .181 | -.274\* |
| 9 | Xob | .086 | -.060 | .536\* | .257\* | .208\* | .187\* | .056 | -.255\* | **.996**\* | .320\* | .285\* | .294\* | -.191 | .474\* | .242\* | .148 | .297\* |
| 10 | Aob | .334\* | .156\* | .261\* | .598\* | .181\* | .213\* | .516\* | .182\* | .353\* | **.995**\* | .267\* | .162\* | -.505\* | .154 | .520\* | .276\* | .037 |
| 11 | Cob | .276\* | .074 | .119\* | .223\* | .705\* | .338\* | .285\* | -.041 | .274\* | .261\* | **.992**\* | .473\* | -.329\* | .043 | .120 | .516\* | .500\* |
| 12 | Oob | .186\* | .099 | .005 | .168\* | .372\* | .672\* | .110\* | -.035 | .267\* | .205\* | .490\* | **.992**\* | -.126 | -.071 | .115 | .184 | .437\* |
| | BFQ-C | | | | | | | | | | | | | | | | | |
| 13 | Neuro | -.409\* | -.010 | -.398\* | -.744\* | -.463\* | -.274\* | -.412\* | -.127 | -.157 | -.533\* | -.343\* | -.161 | **.981**\* | -.235\* | -.429\* | -.367\* | -.234\* |
| 14 | Extra | .140\* | .056 | .701\* | .295\* | .269\* | .204\* | .017 | -.018 | .417\* | .151 | .036 | -.077 | -.205\* | **.953**\* | .423\* | .330\* | .344\* |
| 15 | Agree | .401\* | .236\* | .376\* | .726\* | .344\* | .279\* | .361\* | .110 | .259\* | .503\* | .103 | .110 | -.417\* | .469\* | **.983**\* | .471\* | .326\* |
| 16 | Cons | .370\* | .176\* | .301\* | .491\* | .793\* | .408\* | .308\* | .178 | .148 | .282\* | .501\* | .175 | -.374\* | .345\* | .477\* | **.991**\* | .427\* |
| 17 | Open | .180\* | -.084 | .386\* | .253\* | .549\* | .470\* | .120 | -.254\* | .243\* | .057 | .512\* | .450\* | -.238\* | .315\* | .333\* | .428\* | **.983**\* |

*Note 1.* Sample sizes: correlation within self-report HEXACO-MSI-E: *N* = 1175; correlation between HEXACO-MSI-E and HEXACO-MSI-E Obs: *N* = 343; correlation between HEXACO-MSI-E and BFQ-C: *N* = 400; correlation within HEXACO-MSI-E Obs: *N* = 403; correlation between HEXACO-MSI-E Obs and BFQ-C: *N* = 116; correlation within BFQ-C: *N* = 400.

*Note 2.* ^Correlations between latent factors (below the diagonal) and observed scores (above the diagonal). In the diagonal, correlations between latent factors and observed scores.

*FDR corrected p < .05.

*Note 3.* Results on correlations of observed score within the HEXACO-MSI-E are already published in a previous work [56].

As regards the correlation coefficients between the HEXACO-MSI-E self-report traits and their BFQ-C counterparts, three of them were strong (correlations between HEXACO-MSI-E and BFQ-C X-Extraversion, A-Agreeableness and C-Conscientiousness), one was adequate (O-Openness), one negligeable (between E and Neuroticism). Taking aside this last correlation (and the extra-trait H), 17 out of 20 correlations between different traits assessed by different methods were lower than the above-described correlations. The three higher correlations were due to A correlating strongly and negatively with Neuroticism and moderately and positively with Conscientiousness as well as C correlating moderately with Openness. In general, out of the 20 correlations between different traits assessed by different methods, in absolute value, two are negligible (.00-.09), 12 weak (.10-.39), 5 moderate (.40-.69) and one strong (.70-.89). As mentioned, E does not correlate with Neuroticism and has null or weak correlations with the other BFQ-C factors. As far as the H trait is concerned, it shows weak but consistent co-variation with Neuroticism, Agreeableness and Conscientiousness.

Finally, as expected, correlations between observed and latent dimensions are very strong within each method (min .953), indicating that factor scores are representative of the estimated constructs.

## Criterion validity with respect to self-report and observer criteria

In this section, we will present data on criterion validity of the self-report HEXACO-MSI-E traits with respect to a particular group of criteria connected to: 1) relevant dimensions connected to internalizing and externalizing problems of adolescents (YSR and YOR); 2) risk factors in adolescence (YRB); 3) personal passions, values, beliefs and behaviors (PVB&B); 4) desired features of a possible future job (FFJ). Each group of criteria has been measured in two different forms, self-report and observer (parent or legal guardian), and we have verified how the six traits predict indicators of each group considered in both versions. For each group, zero-order correlations between the criterion indices and the six HEXACO traits of adolescents were preliminarily calculated and are shown in S1 File (see Tables B, C, D and E in S1 File).

**Criterion validity with respect to dimensions connected to internalizing and externalizing problems of adolescents in self-report and observer form.** Correlations between the same items but in self-report (YSR) and observer (YOR) form for specific and general internalizing and externalizing problems ranged between .24 and .47 (N = 125; see Table P in S1 File for each specific correlation).

Two sets of multiple regression analyses with the six self-report HEXACO scales as predictors and each score of the youth emotional and behavioral problems (YSR) as target were performed for both self-report and observer criteria (respectively, Table 3a and 3b). All the models were significant (see Tables F to L in S1 File).

The following factors significantly predicted the following self-report YSR criteria (N = 388): H negatively predicted all the YSR criteria (6 significant regressions coefficients in all) apart from one; E positively all the YSR criteria but three, delinquent behavior is the only one that correlates negatively (4); X negatively predicted all the YSR criteria (7); A negatively predicted only two YSR scores (2); C negatively all the YSR criteria (6) but one; finally, O positively predicted three YSR criteria (3).

The following factors significantly predicted the following YOR criteria (N = 373): H negatively predicted three criteria (3 in all); E negatively one (1); X negatively four (4); A negatively two and one positively (3); C negatively two (2); finally, O negatively one criterion (1).

Significant observer traits predictions (YOR) were few with respect to the self-report analogous (YSR); however, in 11 cases out of 14 they shared the sign.

**Table 3. Descriptive statistics and Stepwise Multiple Regressions with the Self-Report HEXACO-MSI-E traits as predictors and (a) YSR (N = 388) and (b) YOR (N = 373) as target.**

| Youth Emotional and Behavioral Problems | | | | HEXACO-MSI-E | | | | | |
|---|---|---|---|---|---|---|---|---|---|
| **(a) Self-report (YSR)** | **M** | **SD** | **$R_c^2$** | **H** | **E** | **X** | **A** | **C** | **O** |
| *Broad-band scales* | | | | | | | | | |
| Internalizing (20–60) | 33.12 | 8.75 | .64 | -.10* | .21* | -.69* | - | -.12* | .16* |
| Externalizing (27–81) | 37.80 | 7.74 | .55 | -.27* | - | -.12* | -.32* | -.28* | - |
| *Syndrome scales* | | | | | | | | | |
| Anxious/Depressed (10–30) | 17.16 | 4.85 | .59 | -.11* | .32* | -.60* | - | -.12* | .17* |
| Withdrawn/Depressed (7–21) | 11.23 | 3.41 | .59 | - | - | -.76* | - | -.09* | .12* |
| Somatic Complaints (3–9) | 4.73 | 1.55 | .26 | -.23* | .15* | -.41* | - | - | - |
| Delinquent Behavior (11–33) | 13.98 | 2.93 | .37 | -.37* | -.13* | -.17* | - | -.25* | - |
| Aggressive Behavior (16–48) | 23.82 | 5.47 | .55 | -.19* | - | -.11* | -.42* | -.26* | - |
| **(b) Observer (YOR)** | **M** | **SD** | **$R_c^2$** | **H** | **E** | **X** | **A** | **C** | **O** |
| *Broad-band scales* | | | | | | | | | |
| Internalizing (20–60) | 29.03 | 5.92 | .22 | - | - | -.47* | - | - | - |
| Externalizing (27–81) | 33.59 | 4.87 | .16 | -.18* | - | - | -.15* | -.19* | - |
| *Syndrome scales* | | | | | | | | | |
| Anxious/Depressed (10–30) | 15.12 | 3.43 | .15 | - | - | -.39* | - | - | - |
| Withdrawn/Depressed (7–21) | 10.28 | 2.69 | .22 | - | -.12* | -.51* | .11* | - | - |
| Somatic Complaints (3–9) | 3.62 | .85 | .07 | - | - | -.27* | - | - | - |
| Delinquent Behavior (11–33) | 12.26 | 1.35 | .05 | -.20* | - | - | - | - | -.10* |
| Aggressive Behavior (16–48) | 21.33 | 3.96 | .18 | -.15* | - | - | -.21* | -.20* | - |

Note.

* FDR corrected p < .05 within each table.

**Criterion validity with respect to risk factors (YRB) in self-report and observer form.** Correlations between the same items of Youth Risk Behavior (YRB) in self-report and observer form ranged between .17 and .63 ($N_{MIN}$ = 80, $N_{MAX}$ = 124; refer to Table Q in S1 File for the values of each correlation).

Two sets of multiple regression were performed with HEXACO traits of the adolescents as predictors and, respectively, the self-report and observer Youth Risk Behaviors (YRB) as target. All the models were significant (see Table M in S1 File). They are shown in Tables 4 and 5.

The HEXACO traits of adolescents significantly predicted 16 self-report YRBs. H predicted six behaviors: negatively possession of a weapon, use of alcoholic drink and marijuana, self-harming behavior, TV watching, and positively age of the first smoking. E predicted three behaviors: negatively being involved in a physical fight and positively having regular breakfast and sleeping hours per night. X predicted five behaviors: negatively being bullied either at school or online, and self-harming behavior, and positively having breakfast and sleeping hours. A predicted three behaviors: positively wearing a helmet and negatively being in a car with someone drinking alcohol and playing video games or similar. C predicted seven behaviors: positively wearing a helmet, wearing a seat belt on a car, having breakfast and sleeping hours, and negatively physical fights, being electronically bullied and playing video games and similar. Finally, O predicted two behaviors: positively being electronically bullied and age of first smoking.

The HEXACO traits of adolescents significantly predicted 14 observer YRBs. H predicted only two behaviors: negatively alcoholic drinking and positively having breakfast. E predicted

**Table 4. Descriptive statistics and stepwise multiple regressions with the self-report HEXACO-MSI-E traits as predictors and self-report YRB as target (N = 385 if not otherwise indicated).**

| Self-report Youth Risk Behaviors (YRB) | M | SD | %[c] | $R_c^2$ | H | E | X | A | C | O |
|---|---|---|---|---|---|---|---|---|---|---|
| When you ride a bike, skateboard or overboard (etc.), how often do you wear a helmet?[a] (1–5) | 2.15 | 1.39 | - | .24 | - | - | - | .16* | .42* | - |
| How often do you wear a seat belt when riding in a car? (1–5) | 3.48 | 1.32 | - | .08 | - | - | - | - | .29* | - |
| Have you ever ridden in a car driven by someone who had been drinking alcohol? (0–1) | - | - | 91.7 | .02 | - | - | - | -.15* | - | - |
| Have you ever carried a weapon, such as a gun, knife, or club? (0–1) | - | - | 94.3 | .07 | -.26* | - | - | - | - | - |
| Apart from a few sips, have you ever had a whole alcoholic drink? (0–1) | - | - | 90.4 | .08 | -.29* | - | - | - | - | - |
| Have you ever used marijuana for a relatively continuous period? (0–1) | - | - | 98.7 | .01 | -.13* | - | - | - | - | - |
| How old were you when you first tried smoking a cigarette, even just one or two puffs? (1–5) | 4.87 | .55 | - | .04 | .14* | - | - | - | - | .13* |
| In the last 30 days, on the days you smoked, how many cigarettes (regular or electronic) per day have you smoked? (1–6) | 1.02 | .20 | - | - | - | - | - | - | - | - |
| Have you ever engaged in self-harming behaviors (cutting yourself, scratching yourself, etc.) on a voluntary basis? (0–1) | - | - | 87.0 | .07 | -.14* | - | -.24* | - | - | - |
| Have you ever been involved in a physical fight (e.g., you got into a fight, brawls, etc.)? (0–1) | - | - | 72.5 | .11 | - | -.24* | - | - | -.23* | - |
| Have you ever been bullied at school? (0–1) | - | - | 75.8 | .05 | - | - | -.23* | - | - | - |
| Have you ever been electronically bullied (through texting, Instagram, Facebook, or other social media)? (0–1) | - | - | 88.8 | .08 | - | - | -.19* | - | -.23* | .17* |
| During the past 7 days, how many days have you eaten breakfast? (1–8) | 5.91 | 2.52 | - | .06 | - | .11* | .18* | - | .11* | - |
| When you have class, how many hours do you sleep per night? (1–7) | 4.46 | 1.29 | - | .09 | - | .20* | .16* | - | .17* | - |
| On a typical school day (even if in DAD), how many hours a day do you watch TV?[b] (1–6) | 2.73 | 1.54 | - | .04 | -.20* | - | - | - | - | - |
| On a typical school day (even if in DAD), on average, how many hours a day do you play video games or use computer or mobile phone for things not related to school? (1–7) | 4.63 | 1.96 | - | .05 | - | - | - | -.13* | -.17* | - |

Note.

[a] n = 306;

[b] n = 299.

Only for dichotomous variables:

[c] % = percent of '0' (i.e., 'No') category.

* FDR corrected p < .05.

negatively one behavior, being bullied online. X predicted three behaviors: positively wearing a helmet and negatively being bullied either at school and online. A did not predict any risk behavior. C predicted five behaviors: positively wearing a seat belt when riding a car and sleeping hours and negatively being electronically bullied, self-harming behavior and playing video games and similar. Finally, O predicted three behaviors: negatively physical fights and playing video games and positively being electronically bullied.

Traits predicted more self-report than observed risk behaviors. Out of the 14 observed predictions, 8 were shared with self-report ones.

**Criterion validity with respect to passions, values, beliefs and behaviors (PVB&B) in self-report and observer form.** Correlations between the same items of PVB&B in self-report and observer form ranged between .19 and .64 (N = 124; see Table R in S1 File for specific correlations).

Two set of multiple regressions were executed with the six HEXACO traits of adolescents as predictors and each of self-report (N = 382) and observer (N = 373) PVB&B criteria. All the models were significant (see Table N in S1 File). They are shown in Tables 6 and 7.

The six HEXACO traits predicted 38 self-report PVB&B criteria. H predicted seven criteria, four positively (attending religious services, loving animals, respecting older people and using

**Table 5. Descriptive statistics and stepwise multiple regressions with the self-report HEXACO-MSI-E traits as predictors and observer YRB as target (N = 373 if not otherwise indicated).**

| Reported by observer | | | | | HEXACO-MSI-E | | | | | |
|---|---|---|---|---|---|---|---|---|---|---|
| Youth Risk Behaviors (YRB) | $M$ | $SD$ | %[c] | $R_c^2$ | H | E | X | A | C | O |
| When your son/daughter rides a bike, skateboard or overboard (etc.), how often does he/she wear a helmet?[a] (1–5) | 2.16 | 1.47 | - | .05 | - | - | .24* | - | - | - |
| How often does your son/daughter wear a seat belt when riding in a car? (1–5) | 3.61 | 1.37 | - | .02 | - | - | - | - | .14* | - |
| Has he/she ever ridden in a car driven by someone who had been drinking alcohol? (0–1) | - | - | 98.1 | - | - | - | - | - | - | - |
| Has he/she ever carried a weapon, such as a gun, knife, or club? (0–1) | - | - | 99.2 | - | - | - | - | - | - | - |
| Apart from a few sips, has he/she ever had a whole alcoholic drink? (0–1) | - | - | 97.6 | .01 | -.10* | - | - | - | - | - |
| In the last 30 days, on the days he/she smoked, how many cigarettes (regular or electronic) per day has he/she smoked? (1–6) | 1.00 | .00 | - | - | - | - | - | - | - | - |
| Has he/she ever engaged in self-harming behaviors (cutting yourself, scratching yourself, etc.) on a voluntary basis? (0–1) | - | - | 97.6 | .03 | - | - | - | - | -.19* | - |
| Has he/she ever been involved in a physical fight (e.g., he/she got into a fight, brawls, etc.)? (0–1) | - | - | 96.5 | .01 | - | - | - | - | - | -.11* |
| Has he/she ever been bullied at school? (0–1) | - | - | 84.2 | .05 | - | - | -.22* | - | - | - |
| Has he/she ever been electronically bullied (through texting, Instagram, Facebook, or other social media)? (0–1) | - | - | 92.8 | .07 | - | -.11* | -.20* | - | -.16* | .21* |
| During the past 7 days, how many days has your son/daughter eaten breakfast? (1–8) | 6.58 | 2.32 | - | .02 | .14* | - | - | - | - | - |
| When he/she has class, how many hours does your son/daughter sleep per night? (1–7) | 4.93 | .97 | - | .08 | - | - | - | - | .28* | - |
| On a typical school day (even if in distance learning), how many hours a day does your son/daughter watch TV?[b] (1–6) | 2.60 | 1.21 | - | - | - | - | - | - | - | - |
| On a typical school day (even if in distance learning), on average, how many hours a day does your son/daughter play video games or use computer or mobile phone for things not related to school? (1–7) | 4.77 | 1.61 | - | .07 | - | - | - | - | -.20* | -.12* |

Note.

[a] n = 241;

[b] n = 321.

Only for dichotomous variables:

[c] % = percent of '0' (i.e., 'No') category.

* FDR corrected p < .05.

polite words) and three negatively (believing in social network or Internet versus science, keeping a personal diary and composing written arts). E predicted seven criteria, five positively (believing in God, attending religious services, respecting older people, using polite words and keeping a diary) and two negatively (being engaged and playing sports). Out of the nine criteria predicted by X, seven were positive (believing in God, attending religious services, respecting older people, preferring science to art, feeling good, joking and playing sports) and two negative (keeping a personal diary and composing written arts). A predicted four criteria in all, three positively (respecting older people, keeping the promise made, and feeling good) and one negatively (listening to music). C predicted six behaviors, four positively (attending religious services, respecting older people, using polite words and keeping the promise made) and two negatively (being engaged and sleeping late). Finally, O predicted five behaviors, three positively (listening to music, writing a diary, composing written arts) and two negatively (believing social networks or Internet rather than science and feeling good).

The six HEXACO traits predicted 27 observed criteria. H predicted four criteria, two positively (respecting older people and using polite words) and two negatively (believing in social network and Internet rather than in science and playing sports). E predicted five criteria, three positively (believing in God, attending religious services, keeping a diary) and two negatively (preferring science to art, playing sports). X predicted only two behaviours positively

**Table 6. Descriptive statistics and stepwise multiple regressions with the self-report HEXACO-MSI-E traits as predictors and the self-report PVB&B criteria as target (N = 382).**

| Self-report | | | | | HEXACO-MSI-E | | | | | |
|---|---|---|---|---|---|---|---|---|---|---|
| Adolescents' passions, values, beliefs, and behaviors (PVB&B) | M | SD | %[b] | $R_c^2$ | H | E | X | A | C | O |
| I believe in God[a]. | - | - | 15.7 | .06 | - | .17* | .14* | - | - | - |
| I believe that attending religious services (masses, weddings, funerals, etc.) is important. | 3.64 | 1.34 | - | .16 | .14* | .17* | .18* | - | .20* | - |
| I prefer to believe what is said on social networks or on the Internet rather than what science says. | 1.90 | 1.14 | - | .10 | -.17* | - | - | - | - | -.25* |
| I play a musical instrument, or I studied singing[a]. | - | - | 46.9 | - | - | - | - | - | - | - |
| I love animals. | 4.45 | 1.02 | - | .01 | .10* | - | - | - | - | - |
| I prefer science to art. | 3.31 | 1.43 | - | .02 | - | - | .13* | - | - | - |
| I listen to music. | 4.27 | 1.02 | - | .03 | - | - | - | -.12* | - | .15* |
| I write or I wrote a personal diary. | 2.18 | 1.40 | - | .10 | -.16* | .14* | -.11* | - | - | .22* |
| I happened to write poems, stories, or books. | 2.02 | 1.26 | - | .14 | -.14* | - | -.20* | - | - | .36* |
| I play sports. | 3.41 | 1.39 | - | .12 | - | -.12* | .31* | - | - | - |
| I was volunteering. | 1.74 | 1.18 | - | - | - | - | - | - | - | - |
| I feel good. | 4.08 | 1.03 | - | .32 | - | - | .53* | .13* | - | -.09* |
| I make jokes or laugh at other people's jokes. | 4.03 | 1.08 | - | .09 | - | - | .31* | - | - | - |
| I am or I was seriously engaged[a]. | - | - | 80.6 | .04 | - | -.15* | - | - | -.13* | - |
| When there is no school, I sleep late regardless of when I go to bed. | 3.64 | 1.29 | - | .03 | - | - | - | - | -.19* | - |
| I listen to older people with respect. | 4.01 | 1.03 | - | .24 | .18* | .13* | .21* | .15* | .16* | - |
| I say 'please' and 'thank you'. | 4.38 | .96 | - | .17 | .18* | .16* | - | - | .25* | - |
| I try to keep the promises I made. | 4.10 | .92 | - | .18 | - | - | - | .16* | .35* | - |

*Note.* For all criteria, value range is 1–5 (if not otherwise indicated);

[a]value range = 0–1.

Only for dichotomous variables:

[b]% = percent of '0' (i.e., 'No') category.

* FDR corrected p < .05.

(attending religious services and feeling good). A predicted three criteria, one positively (preferring science to art) and two negatively (writing a diary and joking). Out of the nine criteria predicted by C, seven were positive (believing in God, attending religious services, playing an instrument, respecting older people, using polite words, keeping the promises made, playing sports) and two negative (loving animals and sleeping late). Finally, O predicted four criteria, three positive (using polite words, writing a diary, compose written arts) and one negative (believing social network and Internet rather than science).

Traits predicted more self-report (38) than observed (27) PVB&B criteria. Out of the 27 observed predictions, 17 were shared with self-report ones.

**Criterion validity with respect to features of a future job in self-report and observer form.** Correlations between the same items of the features of a possible future job of the participant (FFJ) in self-report and observer form ranged between .18 and .36 (N = 124; see Table S in S1 File for each correlation).

Two sets of regression analyses were performed with the six self-report HEXACO traits as predictors and seven criteria regarding FFJ, evaluated by the participants themselves (self-report; N = 382) and by the parent/legal guardian (observer; N = 373). All the models were significant (see Table O in S1 File). They are shown in (Table 8a and 8b).

The six HEXACO traits predicted 12 self-report FFJ criteria. H predicted positively a job either honest or in contact with the nature and negatively a job that guarantees a good income.

**Table 7. Descriptive statistics and stepwise multiple regressions with the self-report HEXACO-MSI-E traits as predictors and the observer PVB&B criteria as target (N = 373).**

| Reported by observer | | | | | HEXACO-MSI-E | | | | | |
|---|---|---|---|---|---|---|---|---|---|---|
| Adolescents' passions, values, beliefs, and behaviors (PVB&B) | *M* | *SD* | %$^b$ | $R_c^2$ | H | E | X | A | C | O |
| He/She believes in God$^a$. | - | - | 5.1 | .04 | - | .14* | - | - | .13* | - |
| He/She believes that attending religious services (masses, weddings, funerals, etc.) is important. | 3.50 | 1.16 | - | .11 | - | .15* | .14* | - | .22* | - |
| He/She prefers to believe what is said on social networks or on the Internet rather than what science says. | 1.94 | 1.04 | - | .09 | -.15* | - | - | - | - | -.24* |
| He/She plays a musical instrument, or he/she studied singing$^a$. | - | - | 53.9 | .01 | - | - | - | - | .12* | - |
| He/She loves animals. | 4.45 | .88 | - | .01 | - | - | - | - | -.12* | - |
| He/She prefers science to art. | 3.19 | 1.28 | - | .03 | - | -.11* | - | .15* | - | - |
| He/She listens to music. | 4.13 | .90 | - | - | - | - | - | - | - | - |
| He/She writes or wrote a personal diary. | 2.00 | 1.21 | - | .07 | - | .12* | - | -.21* | - | .19* |
| He/She happened to write poems, stories, or books. | 1.94 | 1.08 | - | .13 | - | - | - | - | - | .36* |
| He/She plays sports. | 3.24 | 1.34 | - | .08 | -.11* | -.20* | - | - | .22* | - |
| He/She was volunteering. | 1.29 | .73 | - | - | - | - | - | - | - | - |
| He/She feels good. | 4.56 | .69 | - | .05 | - | - | .23* | - | - | - |
| He/She makes jokes or laugh at other people's jokes. | 4.07 | 1.05 | - | .01 | - | - | - | -.12* | - | - |
| He/She is or was seriously engaged$^a$. | - | - | 97.6 | - | - | - | - | - | - | - |
| When there is no school, He/She sleeps late regardless of when he/she goes to bed. | 3.24 | 1.24 | - | .04 | - | - | - | - | -.20* | - |
| He/She listens to older people with respect. | 4.41 | .83 | - | .14 | .21* | - | - | - | .25* | - |
| He/She says 'please' and 'thank you'. | 4.35 | .85 | - | .09 | .14* | - | - | - | .14* | .13* |
| He/She tries to keep the promises he/she made. | 3.92 | .86 | - | .10 | - | - | - | - | .33* | - |

*Note.* For all criteria: value range is 1–5 (if not otherwise indicated);

$^a$value range = 0–1.

Only for dichotomous variables:

$^b$% = percent of '0' (i.e., 'No') category.

* FDR corrected p < .05.

E predicted positively only a job needed collaboration with other people. X predicted positively a job that allows to collaborate with other people and that is organized and planned and negatively a job connected with art. A predicted positively only a job that allows to be in contact with the nature. C predicted positively an organized and planned job. Finally, O predicted positively a job that provides emotional security, and that allows to be in contact with nature and art.

The six HEXACO traits predicted 12 observer FFJ criteria. H predicted positively an honest job and negatively a job that guarantees a good income. X predicted positively a good income and negatively a job that allows to be in contact with art. A predict a job that allows to collaborate with people. C predicted an organized and planned job. Finally, O predicted a job that allows to be in contact with nature and art.

The six HEXACO traits predicted more self-report (12) than observer (8) features of a future job. Out of the eight predicted observed, six were shared with the self-report ones.

## Discussion and conclusions

In the next sections, we will briefly discuss the most important outcomes of this contribution with respect to the declared aims and delineate possible implications of the results of the research as well as its limitations and future developments.

**Table 8. Descriptive statistics and stepwise multiple regressions with the self-report HEXACO-MSI-E traits as predictors and questions regarding the features of future job (FFJ) as target.**

| Questions regarding a possible future job (FFJ) | $M$ | $SD$ | $R_c^2$ | HEXACO-MSI-E | | | | | |
|---|---|---|---|---|---|---|---|---|---|
| | | | | H | E | X | A | C | O |
| **(a) Self-report (N = 382)** | | | | | | | | | |
| It is important for you that the job is honest. | 4.26 | .90 | .06 | .25* | - | - | - | - | - |
| It is important for you that the job guarantees a good income. | 3.66 | .94 | .04 | -.21* | - | - | - | - | - |
| It is important for you that the job provides emotional security. | 3.85 | .91 | .02 | - | - | - | - | - | .13* |
| It is important for you that the job allows you collaborate with other people. | 3.67 | 1.04 | .05 | - | .14* | .21* | - | - | - |
| It is important for you that the job is organized and planned. | 3.75 | 1.02 | .05 | - | - | .15* | - | .16* | - |
| It is important for you that the job allows you to be in contact with nature. | 2.88 | 1.13 | .07 | .12* | - | - | .12* | - | .16* |
| It is important for you that the job allows you to be in contact with art. | 2.48 | 1.20 | .12 | - | - | -.10* | - | - | .35* |
| **(b) Observer (N = 373)** | | | | | | | | | |
| It is important for your son/daughter that the job is honest. | 4.32 | .80 | .03 | .17* | - | - | - | - | - |
| It is important for your son/daughter that the job guarantees a good income. | 3.80 | .86 | .07 | -.25* | - | .15* | - | - | - |
| It is important for your son/daughter that the job provides emotional security. | 3.89 | .87 | - | - | - | - | - | - | - |
| It is important for your son/daughter that the job allows him/her to collaborate with other people. | 3.63 | .96 | .01 | - | - | - | .13* | - | - |
| It is important for your son/daughter that the job is organized and planned. | 3.54 | .93 | .01 | - | - | - | - | .13* | - |
| It is important for your son/daughter that the job allows him/her to be in contact with nature. | 2.89 | 1.14 | .02 | - | - | - | - | - | .14* |
| It is important for your son/daughter that the job allows him/her to be in contact with art. | 2.57 | 1.16 | .07 | - | - | -.12* | - | - | .27* |

*Note.* For all criteria: range value = 1–5.

* FDR corrected $p < .05$.

## The HEXACO-MSI-E observer

One important outcome is having provided a validation of an Observer Report of the HEXACO-MSI-E that we make public and available to interested researchers. This result can be considered as a relevant contribution especially for the fields of child and adolescent psychology, where reports from significant others are particularly important. Indeed, the factorial structure and the reliability of the Observer Report of the HEXACO-MSI-E was confirmed. Therefore, it is possible to affirm that the observer version of the HEXACO-MSI-E provides the same valid and reliable measure of the adolescent HEXACO traits as the self-report version [56]. However, some correlations between factors from CFA were relatively high and the patterns of correlations in the Observer Report was pretty similar to the ones of self-report version [56]. We hypothesize that the relatively high correlations in the Observer Report could be due to the constraints imposed by the model of CFA on the facets and on the impact of individual differences between parents/legal guardians in responding to the questions about their adolescents in a desirable versus undesiderable way. Moreover, one should note that correlations between latent factors tend to be higher than between observed scale scores, given that in a CFA approach latent correlations are disattenuated for unreliability [60].

## Construct validity

Another important result of the research is having reported strong evidence of construct validity, contrasting the self-report HEXACO-MSI-E inventory, first, with the analogous Observer Report and then, with BFQ-C, meant to measure personality in children from the perspective of FFM.

While construct convergent and divergent validity of the HEXACO-MSI-E with respect to its observer counterpart was excellent and straightforward, correlations between BFQ-C needs

more elaboration. Indeed, all the dimensions converged with their BFQ-C homologue apart from E, and, in general, the dimensions with convergent validity also showed a divergent validity almost satisfying with some interesting 'anomalies'. We want to discuss now these 'anomalies' taking into consideration some important references: the theoretical difference between the two models (the HEXACO and the FFM), the correlations found in adult literature between the two instruments, in particular, the recent meta-analysis by Thielmann et al. [61], on 152 published and unpublished studies, the difference and the similarities between the specific contents of the specific instruments used (HEXACO-MSI-E and BFQ-C), and, finally, the similarities with the analogous correlations between the two instruments when we used the first version of the inventory [55]. We start from the almost null correlation between E and BFQ-C Neuroticism because they are connected. First, in this study E does not correlate with BFQ-C Neuroticism nor, incidentally, they correlated significantly in the very first version of the inventory (r = .14). In that study [55], we hypothesized that it could be due to the restrictiveness of the facets used for E but this is no longer sustainable given that in the present study all the dimensions, including E, are built to be representative of the contents and the facets of each dimension with respect to the original HEXACO model for adults [56]. Indeed, when appreciating the semantic contents, on one side, of the items of the four facets partaking to E and, on the other side, of the BFQ-C Neuroticism scale, it is not unsurprising that the two dimensions are not correlated. The HEXACO one regards mainly fear, dependence, anxiety, and sentimentality while the BFQ-C Neuroticism anger, agitation and nervousness. In sum, it seems reasonable that the two do not correlate given that they reflect very different aspects of personality. However, the facet Patience of A in the HEXACO model is essentially a reversed scale of anger with many of the items expressing rage. Therefore, Neuroticism as measured in BFQ-C and A in the HEXACO model share contents related to rage and anger and, thus, they should correlate negatively. Indeed, the correlation coefficient for A and Neuroticism is significant, strong, and negative (-.744). We observed this pattern also in the very first HEXACO-MSI inventory validated [55]. In sum, notwithstanding that at first sight the low correlation between E and Neuroticism and the strong correlation between A and Neuroticism seem to contradict, respectively, the convergent and discriminant validity of the inventory, these results are indeed coherent with the contents of the respective dimensions and with the different theoretical frameworks.

As stated in the results, apart from the correlation between A and BFQ-C Neuroticism, the highest discriminant correlations were found between A and BFQ-C Conscientiousness, C and BFQ-C Openness and Neuroticism, and, finally, O and BFQ-C Conscientiousness. We are going to provide a rationale for the associations found.

In the 30-item BFQ-C we have utilized in this research, Openness is better defined as Intellect because it expresses the capability of succeeding in a school setting (learning, doing well, being good) while most of the HEXACO-MSI-E items partaking to C refer to effectively organizing and successful doing and monitoring schoolwork and assignments. Therefore, given the emphasis on similar contents applied to school setting, it is no surprise if they share variance, as our results show, and also that they share more variance than the one shared in the adult versions of the two instruments (r = .16) [61]. Analogous considerations on contents explain the negative correlation between C and BFQ-C Neuroticism: this last expresses ease of losing temper and getting angry whereas the Prudence facet of C is just linked to the control and delay of certain aspects related to impulses. In any case, C and FFM's Neuroticism correlated negatively also in adult studies [61]. Even the positive correlation between A and BFQ-C Conscientiousness found in our data is echoed, even if with a lighter intensity, in the adults [61]. Explaining the correlation between O and BFQ-C Conscientiousness, which was found in our data, is more difficult, given that in adults their correlation is almost zero [61] and they

don't seem share items' content. Note that this correlation was also found in the very first version of the inventory [55].

As far as regard the extra factor H, as stated in the results, it correlated negatively with BFQ-C Neuroticism, positively with BFQ-C Agreeableness and Conscientiousness. We should underline that the sign of the three correlations is the same found in adults while the intensity is superimposable for H and Agreeableness (.40 and .47), close for H and Conscientiousness (.37 and .24), and farther for H and Neuroticism (-.41 and -.13). Indeed, H shares variance with FFM's Conscientiousness but also with FFM's Agreeableness [61–63]. Particularly, some of the items of the 30-item BFQ-C Conscientiousness are based on a sense of duty that has a common denominator with the Fairness facet of H of the HEXACO-MSI-E. No surprise that, in the literature on adults, Big Five Agreeableness shares contents with the facets Fairness and Sincerity of H. It is well known that the variance of the FFM Neuroticism and Agreeableness was partitioned into the three HEXACO factors, A, E and H [64].

All in all, if we consider the correlations higher than .30 between cousin traits of HEXACO-MSI-E and BFQ-C found in the present research, nine out of 14 showed the same sign and similar intensity with respect to the ones we found in 2020 [55]. This is probably due to systematic and stable differences between the two instruments rather than lacking construct validity of the HEXACO-MSI-E. Therefore, the coefficients only apparently show a lack of convergent or discriminant validity and are indeed coherently explained on the basis of previous research, contents of the scales, differences between the two models, and previous literature on adults' personality. In sum, notwithstanding the seemingly discordant patterns of correlations, construct validity with respect to FFM was also established.

## Criterion validity with respect to the four different areas of self-report criteria

The third important outcome of this research is having established criterion validity too in the four areas investigated.

As far as the YSR, the most predictive HEXACO factors of many emotional and behavioral youth problems were X, H and C. They all predicted the behavioral problems in a negative way: the more a person was honest and humble, extroverted and conscientious, the less they reported such kind of problems. Therefore, these factors, in the direction of their polarity, seem broad-spectrum protective factors in adolescence. The E dimension was found as a predictive factor as well but, predictably, mainly for internalizing problems: lower E was linked to less anxiety and somatic problems. These results partially confirm what Barbaranelli and colleagues' study [53] found: Extraversion and Conscientiousness were negatively associated with Internalizing problems, and Neuroticism positively associated with Internalizing problems. Finally, to some extent, these findings parallel other reports that dishonest adolescents are more susceptible to developing externalizing problems [29].

As regards the risk factors in the teenage years (self-report YRB), data showed that the stronger predictors were C, H and X. These adolescents' personality traits were related to their participation in risky behaviors in a fairly intuitive pattern. Unconscientious, dishonest, not humble, and introverted adolescents were more likely to participate in health-compromising behaviors than their conscientious, honest, humble, and extroverted peers. In other words, adolescents who were irresponsible, not planning (i.e., unconscientious), unfair, pretentious (i.e., dishonest), shy and unsociable (i.e., introverted) were most susceptible to engage in healthy risky behaviors. These results are consistent with Weller and Tikir [33] findings that low levels of C and H were associated with taking risks in health/safety domain, whereas are

not consistent with the same findings where X was not associated with risks taking in health/safety domain.

The traits more predictive of self-report PVB&B in adolescence were X, H and E. Moderately less predictively resulted C, O and A. Many significant predictions show coherence between the predictive trait and the passions, values, beliefs or behaviors. For example, adolescents high in O preferred science and music and used to write art or diary; adolescents high in C, H and E adopted respectful behaviors. Of course, we also found few predictions that can be regarded as neutral, that is, neither favorable nor contrary to the semantic of the traits significantly associated (e.g., people with low scores of H wrote diary or art; girls and boys with low E's score do not like to play sports).

Most HEXACO traits made coherent predictions with respect to the questions asked to the adolescents participating in the research on their possible future job (self-report FFJ): for example, H predicted positively an honest job and negatively a good income, O predicted job allowing to be in contact with nature and art, X people wanted to collaborate with other people, C people wanted an organized and planned job. Some of the arisen patterns were not coherent (e.g., high level of A were associated with a job in contact with the nature).

In their self-report form, the most predictive factors with respect to all the four categories of criteria, were H, X and C, particularly for the YSR and YRB. For the other criteria (PVB&B and FFJ) also E emerged as predictive. All in all, many of the previsions of the criteria were coherent with the trait involved.

In general, values of the correlations between traits and criteria can depend on same factors that we want to highlight. For example, due to the content of some of the chosen criteria (e.g., "Have you ever used marijuana for a relatively continuous period?"), we expected that many respondents scored to the extreme endpoints of the measurement scale and this is what happened for some items within the criteria (see frequency distribution Tables in S1 File). As it is known [65], this may have attenuated the correlations between the personality scale scores and the criteria. It should also be noted that we focused on the influence of broad, basic personality dimensions on specific classes of criteria, and, in general, the relations between broad bandwidth personality traits and specific criteria may typically be limited in terms of effect sizes [66]. Indeed, among the different chosen criteria, the only ones that can properly be considered psychological concepts are internalizing and externalizing problems, for which indeed personality traits explain much more variance than the other criteria considered. Finally, to control the type I error at the family level, we used the False Discovery Rate correction [59]. Despite the use of this conservative procedure, we stress that the significance threshold may have been affected by the sample size. This all applies also to the following section on the observer criteria.

**Criterion validity with respect to observer criteria.** Here we briefly discuss collectively the results of the criteria based on the answers of a significant other (one parent or the legal guardian of the adolescent).

Correlations between self-report and observer criteria showed, for some items, a good level of covariance; however, many other items showed little covariance. Therefore, parents/legal guardians' ratings not necessarily reflect self-report ratings by their children, highlighting the need for external evaluations in adolescence. In any case, we should consider this result with caution because, for methodological reasons connected to the way we formed the samples of adolescents and parents/guardians, sample sizes for these analyses were relatively small. Future studies should confirm these results.

In general, and within each of the four categories of criteria utilized, the self-report criteria significantly associated with traits in adolescents (i.e., 104) were always more than the analogous observer criteria (i.e., 63). What is more, almost two thirds of the significant ones with

observer criteria coincide with the self-report ones (about 67%). This could be due, of course, to a similarity of methods (i.e., self-report) but also indicates that traits often predict criterion from an inside and outside perspective.

The HEXACO traits provided specific predictions of the YOR criteria (observed), differently by the wide-spectrum predictions of many of the YSR criteria (self-report): indeed, X predicted mainly observerd internalizing problems whereas H, A and C predicted mainly externalizing problems, a pattern just found by Mottola et al. [67] when criteria are observed. About X, this result was also found in Barbaranelli and colleagues' study [53].

HEXACO traits predicted few observed risk behaviors (YRB), the most predictive were C (5 predictions), X and O (both 3) while A did not predict any YRB. Also, for these results, similarities have been found in the same study [67]. Indeed, even in the latter C, X and O were the most predictive traits, but in terms of content there is only little overlap between the two studies. In particular, the only overlaps found concern the role of C in predicting wearing a seat belt on a car, sleeping hours, and self-harming behavior; and for the role of X in predicting being electronically bullied. All the predictions unique to the observer form were protective, that is, the higher the trait, the safer the behavior. The fact that here and in another study [67] observer risk behaviors predicted by the HEXACO-MSI traits were few is probably due to the fact that risk behaviors can be, at least in part, hidden by the adolescents to their parents, given the seriousness of some of them. As far as PVB&B regards, there were some new specific predictions with respect to their self-report form, particularly for C and A. Finally, out of all the predictions arisen with the observer form, only two were specific to it with respect to the self-report ones.

This paper adds significantly to the one on content and dimensional validity, invariance across males and females and across the three classes of middle school, and reliability as internal consistency and 1-year stability [56], providing construct and criterion validity of the HEXACO-MSI-E. It may provide also theoretical developments or clinical applications. For example, it should be underlined that this paper confirms and, in part, clarifies for the first time some of the theoretical differences between the BFF and the HEXACO models in adolescence. Understanding adolescents' personalities with a broader approach than the BFF model allows for a more in-depth and detailed view of adolescents' personalities, providing a more comprehensive overview of their characteristics. Having proven a link between personality traits and many specific behaviors in adolescence opens the field to many applications based on interventions on personality traits of adolescents, such as the prevention of risky behaviors, the improving of wellbeing and health of the adolescents, intervening on emotional problems, improving performance, health or security in school, and so on. Indeed, based on the HEXACO-MSI information, it is possible to develop personalized interventions and support programs for adolescents even based on smartphone use [68]. If low scores emerge in dimensions such as kindness or responsibility, targeted strategies can be designed to reinforce these traits. Finally, considering adolescents' personality traits or facets through HEXACO-MSI-E can offer valuable insights into their inclinations, preferences, and abilities. This can assist them in choosing educational and career paths that align more closely with their characteristics.

This study, as any, has also limitations that indicate future prospects of research. For example, given that the inventory was validated only in Italy, we consider it important to adapt the inventory to different languages and replicate the study to verify the psychometric properties of the inventory in different countries and cultures. Moreover, given that we were from the beginning very concerned about the representativeness of the facets and the dimensions of the inventory, we settled on an extended version of the inventory that would guarantee that representativeness. However, this could, in some conditions, be impractical for its length. Therefore, we think it important to develop a validated short form of the HEXACO-MSI-E that might

come in handy under certain testing conditions and practical constraints. Finally, as suggested by one of the reviewers of this paper, for some items in the Social Self-Esteem scale of Extraversion factor, within the observer report from, i.e., "Others enjoy spending time with her/him", "Nobody likes talking to her/him" and "People like her/him", it would be more appropriate to express them in this way: "S/he believes others enjoy spending time with her/him", "S/he believes nobody likes talking to her/him", and "S/he believes people like her/him". This is because they aim at measuring levels of a person's perception of popularity or attractiveness and not measuring objective levels of it. Therefore, we hope that, in future studies, this inventory may be used in view of these suggestions.

## Supporting information

**S1 File. Supplementary materials.**
(DOCX)

## Author Contributions

**Conceptualization:** Francesca Mottola, Augusto Gnisci, Marco Perugini, Vincenzo Paolo Senese, Ida Sergi.

**Data curation:** Francesca Mottola, Augusto Gnisci, Marco Perugini, Vincenzo Paolo Senese, Ida Sergi.

**Formal analysis:** Francesca Mottola, Augusto Gnisci, Vincenzo Paolo Senese, Ida Sergi.

**Funding acquisition:** Lucia Ariemma, Augusto Gnisci, Roberto Marcone, Vincenzo Paolo Senese, Ida Sergi.

**Investigation:** Lucia Abbamonte, Lucia Ariemma, Roberto Marcone, Andrea Millefiorini, Ida Sergi.

**Methodology:** Francesca Mottola, Lucia Abbamonte, Augusto Gnisci, Marco Perugini, Vincenzo Paolo Senese.

**Project administration:** Augusto Gnisci, Andrea Millefiorini, Ida Sergi.

**Resources:** Augusto Gnisci, Ida Sergi.

**Software:** Andrea Millefiorini.

**Supervision:** Francesca Mottola, Augusto Gnisci, Marco Perugini, Vincenzo Paolo Senese.

**Validation:** Francesca Mottola, Augusto Gnisci, Roberto Marcone, Marco Perugini, Vincenzo Paolo Senese.

**Visualization:** Lucia Ariemma.

**Writing – original draft:** Francesca Mottola, Lucia Abbamonte, Augusto Gnisci, Ida Sergi.

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
