## [Decision Letter · Decision Letter 0]

9 Aug 2023

PONE-D-23-19901Construct and criterion validity of the HEXACO Medium School Inventory Extended (MSI-E)PLOS ONE

Dear Dr. Gnisci,

Thank you for submitting your manuscript to PLOS ONE. After careful consideration, we feel that it has merit but does not fully meet PLOS ONE’s publication criteria as it currently stands. Therefore, we invite you to submit a revised version of the manuscript that addresses the points raised during the review process .As an absolute minimum, column 17 must be added. And just to avoid any mistake, could you go back to your data and check that neuroticism (BFQ-C) was correctly calculated?

We look forward to receiving your revised manuscript.

Kind regards,

Frantisek Sudzina

Academic Editor

PLOS ONE

Journal Requirements:

3. We note you have included a table to which you do not refer in the text of your manuscript. Please ensure that you refer to Table 7 in your text; if accepted, production will need this reference to link the reader to the Table.

Additional Editor Comments:

Reviewers noticed several issues that could be improved, and one column that must be added.

Just to avoid any mistake, could you go back to your data and check that neuroticism (BFQ-C) was correctly calculated?

Personally, I would not omit information that were already provided in the original submission.

Reviewers' comments:

Reviewer's Responses to Questions

**Comments to the Author**

1. Is the manuscript technically sound, and do the data support the conclusions?

Reviewer #1: Yes

Reviewer #2: Yes

2. Has the statistical analysis been performed appropriately and rigorously? 

Reviewer #1: Yes

Reviewer #2: Yes

3. Have the authors made all data underlying the findings in their manuscript fully available?

Reviewer #1: Yes

Reviewer #2: Yes

4. Is the manuscript presented in an intelligible fashion and written in standard English?

Reviewer #1: Yes

Reviewer #2: Yes

5. Review Comments to the Author

Reviewer #1: Review of PONE-D-23-19901, “Construct and criterion validity of the HEXACO Medium School Inventory Extended (MSI-E)”

As its title suggests, this manuscript reports on the validity of a questionnaire for assessing the HEXACO personality characteristics in middle-school students. The research is based on a fairly large sample of adolescents (average age about 12.0 years) for whom self-reports (and in most cases parent reports) were obtained; a variety of outcome variables were also assessed in self-report or (for some of the sample) parent reports.

The results generally showed good psychometric properties for the instrument. For example, the expected factor structure of the instrument was largely recovered, self/parent convergent correlations were generally strong (and self/parent discriminant correlations were generally weak), correlations with a measure of the Big Five were largely consistent with expectations, and correlations with a variety of outcome variables (involving emotional or behavioral problems, values and beliefs, risk behaviors, desired qualities of a future job).

I think that this manuscript provides a great deal of useful information about this rather new instrument, which should be of much value for researchers who study personality in children of middle-school age. I believe it is suitable for publication, but I have several recommendations for revision, all of which I think should be easy to incorporate.

1. It would be useful to report frequency distributions for the items of the various criterion outcomes. In many cases it seems likely that most respondents will report a zero level. If this information won’t fit into the existing tables, it could be given in additional tables, perhaps as a supplement.

2. Also, some mention of these frequency distributions will be warranted in cases where the distribution is highly skewed and/or has a large fraction of responses at the highest or lowest possible value. In general, this will attenuate correlations with the personality scale scores, which presumably will be roughly normally distributed.

3. One difference between the HEXACO-MSI-E and the HEXACO-PI-R is that the HEXACO-MSI-E shows much higher correlations between the scales assessing the C and O dimensions. The authors discuss on page 26 the fact that O correlates with Big Five C in this study, noting the prominence of school-related content in the items of these scales. But the authors could also note the high correlation between C and O within the HEXACO-MSI-E itself, presumably for the same reason.

4. The Discussion section seems to be a very detailed recapitulation of the Results; I think it could be greatly shortened.

5. This is not really a recommendation for revision, but I did notice that the authors emphasize those correlations that reached the threshold of statistical significance, in some cases counting those correlations. There is nothing wrong with this, but it should be kept in mind that this threshold depends on the sample size as well as the effect size. Some statistically significant correlations are rather small, and some correlations that do not reach statistical significance in a medium-size sample might have population values that are non-trivial in size. I wanted to mention this only because the authors appeared to emphasize the statistical significance threshold; again, though, there is not necessarily any revision needed on this point.

6. Points about Information in Tables:

The descriptive statistics for the HEXACO-MSI-E scales (self and observer) should be reported somewhere.

In correlation and regression tables, perhaps the F-statistic results could be deleted, because the F values don’t tell the reader directly about the size of the effect.

For all of the outcome variables for which self-reports and observer reports are available, it would be useful to report the self/observer correlations.

Regarding Table 2, the correlations given are based on different samples or subsamples. It would be useful to show (either in the main text or as a supplement) a version of this table based on only those respondents for whom all of these variables were assessed.

Also for Table 2, I didn’t understand why the correlations between HEXACO-MSI-E scales in self-report were not reported. I might have misunderstood something, but otherwise these should be added.

In Table 2, alpha reliabilities should be reported for the scale scores. (By the way, I don’t think the BFQ-C alphas have been reported anywhere else in the manuscript.)

Also in relation to Table 2, the column for BFQ-C Openness is missing; it would be number 17.

In the title of Table 3, could specify “Self-Report HEXACO-MSI-E”

7. Various Minor Corrections

The abbreviation “Obs” can be useful when referring to the name of the instrument, but when writing a full sentence it’s better to say “Observer Report”. This point applies in several places throughout the manuscript; one example is on page 2, line 438.

In several places the authors use the word “tutor” where I think they mean “guardian” (in English, “tutor” is normally someone who teaches a student in one-on-one sessions, whereas “guardian” is normally someone who takes the role of a parent)

In several places “Consciousness” is used where “Conscientiousness” is intended; this could be fixed with a “find-and-replace”

p. 3, line 65: “has showed” should be “has shown”

pp. 4-5, line 92-93: “Longitudinal studies showed” should be “Longitudinal studies have shown”

p. 5: “scale” is sometimes used in referring to the entire instrument, but “instrument” or “inventory” or “questionnaire” would be better

p. 5, line 117: “provided” should be “provide” (it is used here in the present tense, not the past)

p. 6, line 122: “filled up” should be “filled out” (one fills up a cup or bowl but fills out a questionnaire)

p. 6, line 127: “BFF” has not yet been declared; it could be given as “Big Five” instead

p. 6, line 143: “internalization and externalization problems” are usually stated as “internalizing and externalizing problems”

p. 10, line 233: could specify “low C” for the secondary loading of Dependence on C

p. 14, line 289: could specify as “six self-report HEXACO scales”

p. 14, line 294: “the unique” should be “the only one”

p. 25, line 455: “homologous” should be “counterpart”

p. 25, line 451: “Tielman” should be “Thielmann”

p. 27, line 506: “homo- and hetero-trait” should be “mono- and hetero-trait” (following the conventional terms for the multitrait-multimethod matrix)

p. 28, line 516: “more emotional stability” would probably be better as “lower E” or “less emotionality”, because E is not exactly opposite to emotional stability (as conventionally understood in the Big Five system)

p. 28, line 521: “in teenage age” would sound better as “in the teenage years”

p. 29, lines 548-549: should “YOS” and “YRS” be “YOR” and “YSR”, respectively?

p. 29, line 560: “we have proved evidence” should be “we have provided evidence”

p. 29, line 562: “reconducted” should be “attributed” (I think)

Reviewer #2: The present research reports on psychometric properties of the observer report form of HEXACO-MSI-E as well as on further evidence for construct validity of self-report form of the inventory. The HEXACI-MSI-E was shown to have favorable psychometric properties in terms of factor structure, internal consistency reliability, and criterion-related validity. Given the importance of parents/caretakers’ reports in adolescent psychology, developing and validating the observer report form of the HEXACO-MSI-E seems to make an important contribution to the literature.

Below, I provide some suggestions that the authors may consider in revising the current manuscript.

1. On p. 12, the relationships between the BFQ-C and HEXACO-MSI-E variables are described using the MTMM terms (e.g., homotrait-heteromethod coefficients). As the authors mentioned in the discussion section, the FFM variables should not be considered the same as the HEXACO variables. In this sense, it may be incorrect to call some of the correlations involving the BFQ-C variables to be homotrait-heteromethod coefficientst; as such, I would remove BFQ-C variables from the MTMM matrix (Table 2). Relationships between the HEXACO-MSI-E and BFQ-C can be described in another table (i.e., 5 BFQ-C by 6 self-report of the HEXACO-MSI-E). This way, the authors might be able to streamline the results presentation on page 12 and the associated discussions on pages 25—27.

2. In Table 2, I wasn’t sure why the values above the diagonal for self-reports HEXACO-MSI-E variables are all blank.

3. In Table 2, column 17 is missing.

4. It would be helpful to mention somewhere in the text (and also in Tables 3, 4, 5, 6, and 7) that self-reports of the HEXACO-MSI-E were used in these analyses (as opposed to observer reports). (By the way, am I correct on this?)

5. I have some suggestions for Tables 3, 4, 5, 6, and 7, which the authors may or may not accept: (1) I would omit F values and dfs from the tables for the sake of simplicity. (2) I am wondering if it is more straightforward to conduct standard multiple regression analyses instead of stepwise multiple regression analyses. (3) it might be informative to provide zero-order correlations in addition to (or instead of) beta coefficients.

6. The discussions involving criterion-related validity (from p. 27) seem to be a bit repetitive to what was stated in the results section, and it might be helpful if the authors could streamline it by discussing some implications of the results.

7. I have one suggestions for some items in the Social Self-Esteem scale within the observer report from; e.g., “Others enjoy spending time with her“, “Nobody likes talking to her” and “People like her”. These items are not intended to measure objective levels of the target person’s popularity or attractiveness, but to measure target person’s perception of it. As such, it might be more appropriate to add “S/he believes” or “S/he thinks” to those items.

8. The manuscript may benefit from a more thorough proofreading. The followings are what I spotted while reading the manuscript, and they are not meant to be a thorough list of typos.

Line 65, “showed” “shown”

Line 66, “to a lesser extent” may sound better than “to little extent”

Line 127, “BFF” may need to be spelled out.

Line 160, “had never access” � “never had access”?

Line 187, “in literature is known as” � In the literature it is known as”?

Throughout, “As regard” should be “As regards”

Line 461, “Tielman” � “Thielmann”

Line 560, “we have proved evidence” � “we have provided evidence”?

6. PLOS authors have the option to publish the peer review history of their article (what does this mean?). If published, this will include your full peer review and any attached files.

Reviewer #1: No

Reviewer #2: No

---

## [Author Response · Author response to Decision Letter 0]

20 Sep 2023

Reviewer #1: Review of PONE-D-23-19901, “Construct and criterion validity of the HEXACO Medium School Inventory Extended (MSI-E)”

As its title suggests, this manuscript reports on the validity of a questionnaire for assessing the HEXACO personality characteristics in middle-school students. The research is based on a fairly large sample of adolescents (average age about 12.0 years) for whom self-reports (and in most cases parent reports) were obtained; a variety of outcome variables were also assessed in self-report or (for some of the sample) parent reports.

The results generally showed good psychometric properties for the instrument. For example, the expected factor structure of the instrument was largely recovered, self/parent convergent correlations were generally strong (and self/parent discriminant correlations were generally weak), correlations with a measure of the Big Five were largely consistent with expectations, and correlations with a variety of outcome variables (involving emotional or behavioral problems, values and beliefs, risk behaviors, desired qualities of a future job).

I think that this manuscript provides a great deal of useful information about this rather new instrument, which should be of much value for researchers who study personality in children of middle-school age. I believe it is suitable for publication, but I have several recommendations for revision, all of which I think should be easy to incorporate.

A: We thank R1 for his/her words and for the provided suggestions that improved the paper very much. As detailed below, we have followed almost all the suggestions by R1.

1. It would be useful to report frequency distributions for the items of the various criterion outcomes. In many cases it seems likely that most respondents will report a zero level. If this information won’t fit into the existing tables, it could be given in additional tables, perhaps as a supplement.

A: We thank R1 for this suggestion. We have reported frequency distributions for the items of the various criterion outcomes in additional tables (Tables F, G, H, I, J, K, L, M, N, and O of Supplementary Materials).

2. Also, some mention of these frequency distributions will be warranted in cases where the distribution is highly skewed and/or has a large fraction of responses at the highest or lowest possible value. In general, this will attenuate correlations with the personality scale scores, which presumably will be roughly normally distributed.

A: We thank R1 for this good point. We have mentioned this aspect in the conclusions of our paper (lines 1120-1123).

3. One difference between the HEXACO-MSI-E and the HEXACO-PI-R is that the HEXACO-MSI-E shows much higher correlations between the scales assessing the C and O dimensions. The authors discuss on page 26 the fact that O correlates with Big Five C in this study, noting the prominence of school-related content in the items of these scales. But the authors could also note the high correlation between C and O within the HEXACO-MSI-E itself, presumably for the same reason.

A: As we reported in the paper, the BFQ-C Openness is better defined as Intellect because it expresses the capability of succeeding in a school setting (learning, doing well, being good). 

In the HEXACO-MSI-E, Openness to Experience is better defined as the attraction toward beauty of art and nature, inquisitiveness about various domains of knowledge and about ideas that may seem radical or unconventional, while most of the HEXACO-MSI-E items partaking to C refer to effectively organizing and successful doing and monitoring schoolwork and assignments. For this reason, we think that it is difficult to explain the high correlation between C and O within the HEXACO-MSI-E in the same way as we explained the high correlation between C and BFQ-C Openness.

4. The Discussion section seems to be a very detailed recapitulation of the Results; I think it could be greatly shortened.

A: We thank R1 for this suggestion. We have streamlined the Discussion, avoiding repetitions, and provided more keys to understanding results and their implications, also based on the few existing literature. 

5. This is not really a recommendation for revision, but I did notice that the authors emphasize those correlations that reached the threshold of statistical significance, in some cases counting those correlations. There is nothing wrong with this, but it should be kept in mind that this threshold depends on the sample size as well as the effect size. Some statistically significant correlations are rather small, and some correlations that do not reach statistical significance in a medium-size sample might have population values that are non-trivial in size. I wanted to mention this only because the authors appeared to emphasize the statistical significance threshold; again, though, there is not necessarily any revision needed on this point.

A: We thank R1 for this observation. In the revised conclusions of our paper (lines 1128-1129) we have highlighted that despite the use of a conservative procedure such as False Discovery Rate correction, the statistical significance threshold may have been affected by the sample size. 

6. Points about Information in Tables:

The descriptive statistics for the HEXACO-MSI-E scales (self and observer) should be reported somewhere.

A: As correctly noted by R1, we did not report descriptive statistics in the first version of the paper. Now, in the Table 1, we have added descriptive statistics for the Obs HEXACO-MSI-E. The descriptive statistics for the self HEXACO-MSI-E are reported in the Table 7 of the already published work of Gnisci et al. (2023).

In correlation and regression tables, perhaps the F-statistic results could be deleted, because the F values don’t tell the reader directly about the size of the effect.

A: We thank you for this suggestion. We have moved the F-statistic results from regression tables in the manuscript to the tables of frequency distribution in the Supplementary Materials. 

For all of the outcome variables for which self-reports and observer reports are available, it would be useful to report the self/observer correlations.

A: This is a very important information to be provided. Unfortunately, for methodological reasons due to the way we sampled the adolescents and the parents/legal guardians (see Method), we have relatively few participants who had self-report and observer ratings. Normally, they were 124-125 but, in some cases, even 80. Therefore, we decided to provide in the text, for each of the four groups of criteria, the minimum and maximum value of the correlations (and N), providing the detailed tables of correlations in the Supplementary Materials. 

Regarding Table 2, the correlations given are based on different samples or subsamples. It would be useful to show (either in the main text or as a supplement) a version of this table based on only those respondents for whom all of these variables were assessed.

A: We thank R1 for this suggestion. We prefer not to apply it because we have too few respondents, which would make the results unreliable. However, we have realized that we did not clearly report subsample sizes on which we computed correlations. We have added this information in a note of Table 2.

Also for Table 2, I didn’t understand why the correlations between HEXACO-MSI-E scales in self-report were not reported. I might have misunderstood something, but otherwise these should be added.

A: In the first version of the paper, we decided to not report the correlations between HEXACO-MSI-E scales in self-report because these had already been reported and published in a previous work (Gnisci et al., 2023). However, as also suggested by the Editor and by R2, as well as by R1, in this new version of the paper, we have added these data and specified that they are already published in a previous work (see note 2 of Table 2). 

In Table 2, alpha reliabilities should be reported for the scale scores. (By the way, I don’t think the BFQ-C alphas have been reported anywhere else in the manuscript.)

A: We thank R1 for this observation. The alpha coefficients for self HEXACO-MSI-E were just published in Gnisci et al. (2023) while the ones for Obs HEXACO-MSI-E are in the Table 1 of the present paper. As far BFQ-C is concerned, in the paragraph "Big Five Questionnaire-Children" of Measures section, we have now added information on the factorial structure of the BFQ-C and the alphas for each of the BFQ-C factors.

Also in relation to Table 2, the column for BFQ-C Openness is missing; it would be number 17.

A: We thank R1 for noting this, we have added the column for BFQ-C Openness (column 17) in the Table 2 and the related correlations.

In the title of Table 3, could specify “Self-Report HEXACO-MSI-E”

A: Done. Thanks.

7. Various Minor Corrections

The abbreviation “Obs” can be useful when referring to the name of the instrument, but when writing a full sentence it’s better to say “Observer Report”. This point applies in several places throughout the manuscript; one example is on page 2, line 438.

A: We thank R1 for this consideration that we have applied in all pertinent places throughout the manuscript.

In several places the authors use the word “tutor” where I think they mean “guardian” (in English, “tutor” is normally someone who teaches a student in one-on-one sessions, whereas “guardian” is normally someone who takes the role of a parent)

A: We thank R1 for this specification, we replaced the word “tutor” with “guardian” or “legal guardian”, depending on the case, in all the manuscript.

In several places “Consciousness” is used where “Conscientiousness” is intended; this could be fixed with a “find-and-replace”

A: Done. Thanks. 

p. 3, line 65: “has showed” should be “has shown”

A: Done.

pp. 4-5, line 92-93: “Longitudinal studies showed” should be “Longitudinal studies have shown”

A: Done.

p. 5: “scale” is sometimes used in referring to the entire instrument, but “instrument” or “inventory” or “questionnaire” would be better

A: We thank R1 for this suggestion. In all the manuscript, we have replaced the term “scale” with “inventory” when referring to entire instrument. 

p. 5, line 117: “provided” should be “provide” (it is used here in the present tense, not the past)

A: Done. Thanks. 

p. 6, line 122: “filled up” should be “filled out” (one fills up a cup or bowl but fills out a questionnaire)

A: Done. 

p. 6, line 127: “BFF” has not yet been declared; it could be given as “Big Five” instead

A: We have removed and replaced it with “Big Five”. Thanks for this observation. 

p. 6, line 143: “internalization and externalization problems” are usually stated as “internalizing and externalizing problems”

A: We have replaced “internalization and externalization problems” with “internalizing and externalizing problems” in all the manuscript. Thanks. 

p. 10, line 233: could specify “low C” for the secondary loading of Dependence on C

A: Done.

p. 14, line 289: could specify as “six self-report HEXACO scales”

A: Done.

p. 14, line 294: “the unique” should be “the only one”

A: Done.

p. 25, line 455: “homologous” should be “counterpart”

A: Done. 

p. 25, line 451: “Tielman” should be “Thielmann”

A: Done. 

p. 27, line 506: “homo- and hetero-trait” should be “mono- and hetero-trait” (following the conventional terms for the multitrait-multimethod matrix)

A: Done.

p. 28, line 516: “more emotional stability” would probably be better as “lower E” or “less emotionality”, because E is not exactly opposite to emotional stability (as conventionally understood in the Big Five system)

A: We thank R1 for this suggestion. We have replaced “more emotional stability” with “lower E”. 

p. 28, line 521: “in teenage age” would sound better as “in the teenage years”

A: Done.

p. 29, lines 548-549: should “YOS” and “YRS” be “YOR” and “YSR”, respectively?

A: We thank R1 for noticing this error. We have replaced “YOS” and “YRS” with “YOR” and “YSR” respectively. 

p. 29, line 560: “we have proved evidence” should be “we have provided evidence”

A: Done. Thanks.

p. 29, line 562: “reconducted” should be “attributed” (I think)

A: We thank R1 for this suggestion. We have replaced “reconducted” with “attributed”.

Reviewer #2: The present research reports on psychometric properties of the observer report form of HEXACO-MSI-E as well as on further evidence for construct validity of self-report form of the inventory. The HEXACI-MSI-E was shown to have favorable psychometric properties in terms of factor structure, internal consistency reliability, and criterion-related validity. Given the importance of parents/caretakers’ reports in adolescent psychology, developing and validating the observer report form of the HEXACO-MSI-E seems to make an important contribution to the literature.

Below, I provide some suggestions that the authors may consider in revising the current manuscript.

A: We thank R2 for his/her words and for the provided suggestions that improved the paper very much. Based on his/her positive consideration, we have emphasized the importance of having a validated observer report form, especially in child and adolescent psychology, in the definition of aims (lines 139-141) and in the discussions of our paper (lines 956-957). As detailed below, we have followed almost all the suggestions by R2.

1. On p. 12, the relationships between the BFQ-C and HEXACO-MSI-E variables are described using the MTMM terms (e.g., homotrait-heteromethod coefficients). As the authors mentioned in the discussion section, the FFM variables should not be considered the same as the HEXACO variables. In this sense, it may be incorrect to call some of the correlations involving the BFQ-C variables to be homotrait-heteromethod coefficientst; as such, I would remove BFQ-C variables from the MTMM matrix (Table 2). Relationships between the HEXACO-MSI-E and BFQ-C can be described in another table (i.e., 5 BFQ-C by 6 self-report of the HEXACO-MSI-E). This way, the authors might be able to streamline the results presentation on page 12 and the associated discussions on pages 25—27.

A: We thank R2 for this point. At the beginning of this study, we were not yet aware of the systematic differences between the BFQ-C and the HEXACO-MSI-E, indeed they emerged later just in the present study. Therefore, we still consider it appropriate to talk about convergent and divergent validity with respect to the BFQ-C but, when describing these results, we abandoned the terminology associated with the MTMM matrix, as R2 suggests. In particular, we specified this aspect in the research questions (“The BFQ-C has approximately a similar behavior domain of HEXACO-MSI-E but rather different contents depending on each trait. Therefore, a priori, we do not expect the traditional results of the multitrait-multimethod matrix whereby the correlations between the ‘same’ traits have to be high and higher than correlations between different traits”. In the Results and Conclusions, we have replaced, for example, "homotrait-heteromethod coefficients" with the term "counterparts" (Line 351) and "heterotrait-heteromethod" with the periphrasis "between different traits assessed by different methods" (Lines 354 and 357). Having adopted these solutions, we think Table 2 can remain in the manuscript as it is because it is very informative.

2. In Table 2, I wasn’t sure why the values above the diagonal for self-reports HEXACO-MSI-E variables are all blank.

A: It is because these data are already reported and published in a previous work. Despite this, in the revised version of paper, following the Editor and R1’s suggestions, we have decided to report the results on correlations of observed score within the self-reports HEXACO-MSI-E specifying in the note 2 of Table 2 that those data have already been published in a previous work and also indicating the reference of the latter.

3. In Table 2, column 17 is missing.

A: We thank R2 for rising up this point. Also following the Editor and R1’s suggestions, we have added, in the Table 2, the column 17 for BFQ-C Openness.

4. It would be helpful to mention somewhere in the text (and also in Tables 3, 4, 5, 6, and 7) that self-reports of the HEXACO-MSI-E were used in these analyses (as opposed to observer reports). (By the way, am I correct on this?)

A: We thank R2 for noticing this missing information. First of all, we confirm that R2 is right; in these analyses, we used the self-reports of the HEXACO-MSI-E. Therefore, following his/her suggestion, we have better specified in different places of the manuscript and also in the captions of Tables 3, 4, 5, 6, 7, and 8 that we used the self-reports of the HEXACO-MSI-E. Specifically, in the text, we specified this aspect in the following paragraphs:

- "Data analysis" (Lines 309 and 310, page 10);

- "Criterion Validity with respect to Self-Report and Observer Criteria" (Line 468, page 14);

- “Criterion Validity with respect to Dimensions Connected to Internalizing and Externalizing Problems of Adolescents in Self-report and Observer Form”(Line 477; page 14);

- "Criterion Validity with respect to Features of a Future Job in Self-report and Observer form" (Line 816, page 23).

5. I have some suggestions for Tables 3, 4, 5, 6, and 7, which the authors may or may not accept: (1) I would omit F values and dfs from the tables for the sake of simplicity. (2) I am wondering if it is more straightforward to conduct standard multiple regression analyses instead of stepwise multiple regression analyses. (3) it might be informative to provide zero-order correlations in addition to (or instead of) beta coefficients.

A: We thank R2 for all the suggestions. We answer below for each point.

1) Following R1’s suggestion, we have removed F values and dfs from regression tables and we have reported them in frequency distribution tables of Supplementary Materials (Tables F to O).

2) Results are not much different. We ran a standard regression (with beta not corrected for multiple testing) on each of the 92 criteria, as R2 suggests, and out of 166 significant correlations that emerged with stepwise in our study, 145 were confirmed and 11 new emerged. Given these similar patterns, we confirmed the stepwise given that it makes possible comparisons with other studies in literature which used also stepwise method.

3) For the sake of simplicity, we have reported the zero-order correlations in Supplementary Materials (Tables B, C, D, E).

6. The discussions involving criterion-related validity (from p. 27) seem to be a bit repetitive to what was stated in the results section, and it might be helpful if the authors could streamline it by discussing some implications of the results.

A: We thank R2 for this suggestion. We have streamlined the discussions involving criterion-related validity and, in general, the whole discussions section, avoiding repetitions. We added more interpretations of the results based on the small existing literature and some theoretical and practical implications of the results.

7. I have one suggestions for some items in the Social Self-Esteem scale within the observer report from; e.g., “Others enjoy spending time with her“, “Nobody likes talking to her” and “People like her”. These items are not intended to measure objective levels of the target person’s popularity or attractiveness, but to measure target person’s perception of it. As such, it might be more appropriate to add “S/he believes” or “S/he thinks” to those items.

A: We thank R2 for this interesting suggestion. In the revised version of this paper, we have included this aspect as a limitation (lines 1165-1171) of our study suggesting the use of this inventory, in future studies, in the light of these tips. To this end, we have also uploaded on our webpage a revised version of Obs HEXACO-MSI-E with a note specifying that, after the validation of the inventory, we realized that three items of the Social Self-Esteem scale would be better expressed in the format that includes "S/he believes".

8. The manuscript may benefit from a more thorough proofreading. The followings are what I spotted while reading the manuscript, and they are not meant to be a thorough list of typos.

Line 65, “showed” “shown”

A: Done.

Line 66, “to a lesser extent” may sound better than “to little extent”

A: Done. 

Line 127, “BFF” may need to be spelled out.

A: We thank R2 for noticing this. Also following R1’s suggestion, we have removed “BFF” and replaced it with “Big Five”.

Line 160, “had never access” � “never had access”?

A: Done.

Line 187, “in literature is known as” � In the literature it is known as”?

Throughout, “As regard” should be “As regards”

A: Done.

Line 461, “Tielman” � “Thielmann”

A: Done. Thanks.

Line 560, “we have proved evidence” � “we have provided evidence”?

A: Done.

6. PLOS authors have the option to publish the peer review history of their article (what does this mean?). If published, this will include your full peer review and any attached files.

Do you want your identity to be public for this peer review? For information about this choice, including consent withdrawal, please see our Privacy Policy.

Reviewer #1: No

Reviewer #2: No

---

## [Editor Report · Decision Letter 1]

28 Sep 2023

Construct and criterion validity of the HEXACO Medium School Inventory Extended (MSI-E)

PONE-D-23-19901R1

Dear Dr. Gnisci,

We’re pleased to inform you that your manuscript has been judged scientifically suitable for publication and will be formally accepted for publication once it meets all outstanding technical requirements.

Kind regards,

Frantisek Sudzina

Academic Editor

PLOS ONE
---

## [Editor Report · Acceptance letter]

6 Oct 2023

PONE-D-23-19901R1 

Construct and criterion validity of the HEXACO Medium School Inventory Extended (MSI-E) 

Dear Dr. Gnisci:

I'm pleased to inform you that your manuscript has been deemed suitable for publication in PLOS ONE. Congratulations! Your manuscript is now with our production department. 

Kind regards, 

on behalf of

Dr. Frantisek Sudzina 

Academic Editor

PLOS ONE